# Brain anatomy of the Cambrian fossil *Jianfengia multisegmentalis* informs euarthropod phylogeny

Nicholas J. Strausfeld [1], David R. Andrew [2], Xianguang Hou[3] & Frank Hirth [4]

Cambrian fossils from the Chengjiang biota demonstrate that over half a billion years ago early stem euarthropods existed coevally with representatives of already recognizable crown groups. Prominent stem taxa were *Fuxianhuia protensa* and *Alalcomenaeus* whose cerebral and ganglionic traits identify them as, respectively, stem mandibulates and stem chelicerates. Here we report on the visual systems and brain of the enigmatic lower Cambrian euarthropod *Jianfengia multisegmentalis*, which reveals neural traits suggestive of Pancrustacea despite its possession of 'great appendages'. As occur in pancrustaceans, three nested optic neuropils are resolved in the eyestalks of *Jianfengia*, together with rostral ocelli and their associated nerves supplying a discrete forebrain region. Sutured eyestalks typifying crown Malacostraca provide compound eyes populated by ommatidia revealing structures suggesting cone-building cells. These and other neuroanatomical traits provide a powerful tool for resolving euarthropod relationships. Phylogenetic analyses deploying neural traits of *Jianfengia*, other Cambrian taxa, and extant Euarthropoda elucidate the status of *Jianfengia* as sister to total Mandibulata and reveal the short-bodied 'great appendage' Leanchoiliidae as sister to total Chelicerata. Together these data provide independent evidence for a 23 year-old proposition that 'great appendage' morphology defines the early stem from which derived the two branches of the euarthropod tree of life.

Fossils of lower Cambrian stem euarthropods exhibit neuromorphological characters that typify the cerebra of crown Euarthropoda[1–3]. Examples include the organization of the optic tracts and neuropils subtending the compound eyes of *Fuxianhuia protensa* that correspond to visual centers defining certain mandibulates. Organization of the fuxianhuiid deutocerebrum supplying nerves to its antenniform appendages corresponds to mandibulate olfactory lobes and antennules. The cerebrum and visual pathways of the megacheiran *Alalcomenaeus* further reveal a cerebral organization corresponding to that of larval *Limulus* (Merostomata)[2]. Such distinctions relate directly to variations of neural organization within each of the three domains of the cerebrum that in extant euarthropods are genetically determined by the combinatorial activity of conserved homeobox transcription factors, thereby indicating their ancient origin[4]. Specific phenotypic distinctions of the internal organization of each cerebral domain further discriminate the euarthropod clades Chelicerata, Arachnida, and

[1]Department of Neuroscience, University of Arizona, Tucson, AZ 85721, USA. [2]Department of Biology, Lycoming College, Williamsport, PA 17701, USA. [3]Yunnan Key Laboratory for Palaeobiology, Institute of Paleontology, Yunnan University, Kunming, China. [4]Department of Basic and Clinical Neuroscience, Institute of Psychiatry, Psychology and Neuroscience, King's College London, London SE5 9RT, UK. e-mail: flybrain@arizona.edu; andrew@lycoming.edu; xghou@ynu.edu.cn; frank.hirth@kcl.ac.uk

**Fig. 1 | Jiangfengia multisegmentalis and its visual systems. a** Optical photograph of of YKLP11117. Its trunk composed of 27 isomorphic segments terminates in a blade-like telson (TE). The 2 mm scale bar lower right demonstrates the minuteness of this taxon. The framed area to the right, which includes the carapace and cerebral elements, is enlarged in (**b**). Within the frame, the upper box indicates the ocellar nature of the anterior sclerite enlarged in Fig. 2a–c, the lower box indicates the right eyestalk enlarged in (**c**, **d**). **b** The three rostral-most segments (T1-T3) and the asegmental cephalon are covered by the carapace (CA) extending forward from the anterior margin of the trunk's (TR) 4th trunk segment (T4). Eyestalks situated behind the paired anterior projections (AP) emerge laterally from the front of the carapace and terminate as compound eyes (right eye RCE, fully exposed, left eye (LCE) mostly buried in the matrix). The midline 'anterior sclerite' carries the ocelli (OC, see Fig. 2a-c). **c, d** Optical photograph with white light and ultraviolet of the enlarged right eyestalk. In **c** oblique illumination from the right accentuates surface features of the compound retina which is enlarged in the inset lower left. In **d** mixed white light and ultraviolet reveal facets with missing lenses (center inset) allowing resolution of underlying potential cone cells, as shown in Supplementary Fig. 2. The inset lower right schematic demonstrates the hexagonal patterning of ommatidia. **e** White light optical photograph of the allied multisegmented great appendage euarthropod *Fortiforceps* showing evidence of a distinct suture (arrowed). Scale bars: *a* = 2 mm; *b* = 1 mm; **c–e** = 50 μm.

Myriapoda. Unrecognized until now are fossilized species that possessed cerebral traits corresponding to those that contribute to defining Pancrustacea, today's most species-rich panarthropod group[5].

Here we describe observations of six specimens of the genus *Jianfengia multisegmentalis* (Hou 1987) retrieved from the Cambrian (Series 2, Stage 3) Eoredlichia–Wutingaspis trilobite biozone, Yu'anshan Member, Chiungchussu Formation[6]. *Jianfengia* is a minute 'great appendage'[7] euarthropod (Fig. 1a), significantly smaller than other fossil taxa in which well-defined neural traces have been identified. To put this in perspective, the cerebrum of the mandibulate stem taxon *Fuxianhuia protensa*[1] is at least three times broader than the cerebrum of *Jianfengia*. Accordingly, neural traces in *Jianfengia* are themselves microscopic and are in some specimens indicated by dense granular deposits rather than continuous dark profiles. We have here deployed strategies that utilize granularity to provide clear evidence of fossilized neuropil thereby providing empirical support for a more traditional approach of neuromorphological reconstruction. Together with phylogenetic analyses, our findings identify *Jianfengia* as a stem euarthropod sister to total Mandibulata exhibiting cerebral traits typifying Malacostraca and Branchiopoda that distinguish it from a cerebral organization defining the 'great appendage' Leanchoiliidae[2,3].

## Results

### External attributes and diagnostic traits

Two *Jianfengia* specimens of similar size, YKLP11117 (Fig. 1a–e) and NIGPAS 100123b (Supplementary Fig. 1), provide views of the external morphology of this species, with an emphasis on its rostral carapace and its associated skeletal attributes. *Jianfengia multisegmentalis* has a segmented trunk, approximately 2.5 cm in length comprising 27-28 homonomous segments (Fig. 1a). A rostral carapace 5 mm in length and barely 3 mm in width covers the cerebrum and the first three post-cerebral segments (Supplementary Table 1). Paired eyestalks extend laterally from the carapace immediately in front of a pair of uniramous postocular 'great appendages' that originate from beneath the carapace (Supplementary Fig. 1a–c). Each 'great appendage' comprises six podomeres of which the second and third provide an elbow-like articulation[7]. The elongated shaft of the third podomere terminates as three articulating blades that together furnish a stubby chela. These appendicular attributes conform to the general category of 'great appendages' ascribed to members of the paraphyletic clade Megacheira (Supplementary Table. 2), the significance of which is discussed later.

Rostrally, a pair of protrusions extending from the front of the carapace flank a substantial forward-projecting anterior sclerite, which arises from beneath the edge of the carapace with which it likely articulates (Figs. 1b, 2a–c). Despite considerable flattening typical of Chengjiang fossils, features of the anterior sclerite indicate that it is capped by three areas we interpret as ocellar-like lenses comparable to those described for the anterior sclerites of the deuteropodian *Odaraia alata*[8], thereby corresponding to frontal eyes of trilobites[9] and the nauplius eye/ocelli of extant pancrustaceans[10,11]. As shown in Fig. 2a-c, these putative ocelli overlie a palisade of rod-like components here interpreted as photoreceptors at the base of the anterior sclerite. Superimposition of neural traces from the anterior sclerite of *Jianfengia* specimen YKLP17299 onto the corresponding exoskeleton of YKLP11117 (Fig. 2d) further supports the presence of ocelli represented by a system of diverging axons from the anterior sclerite. These extend into the most rostral neuropil of the brain in a manner identical to

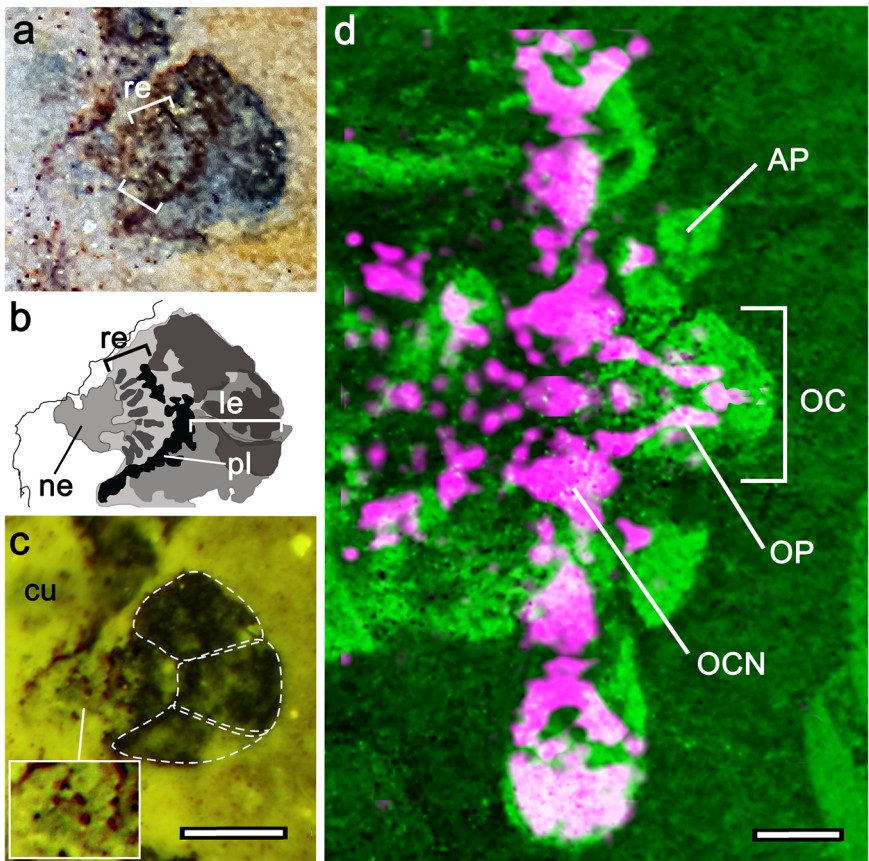

**Fig. 2 | Morphology of the anterior sclerite indicates an ocellar/nauplius-like "eye" in *Jianfengia*. a**, **b** Optical photograph and tracing of YKLP11117's midline anterior sclerite. Despite the compression of the fossil specimen, it reveals concentric arrangements indicating outer dioptric elements interpreted as a triplet of lenses (le) overlying a pigmented layer (pl) beneath which are short columnar elements here interpreted as photoreceptors (re). The entire sclerite arises from a short neck (ne). **c** UV illumination resolves the neck of the sclerite as comprising fold-like elements suggesting unsclerotized arthrodial membrane (inset lower left) extending from beneath the overlying frontal cuticle (cu). **d** Optical photograph of specimen YKLP17299 (rendered as monochrome magenta) superimposed on YKLP11117 (monochrome green) to demonstrate the correspondence between neural features and the external morphology of *Jianfengia*. Ocellar neural pathways (OP) extending to the prosocerebral ocellar neuropils (OCN) indicate that the ocellus (OC) is a bona fide sensory organ supplying the prosocerebrum. Scale bars are 0.25 mm.

axonal projections from a pancrustacean's ocelli into its rostral cerebrum[10,11].

Proven by developmental genetics of extant pancrustaceans, the combinatorial activity of *Six3, FoxQ2* and *hbn* gene homologs is required for the development of rostral ocellar photoreceptors whose axons project into the prosocerebrum of the forebrain[12–14]. The corresponding organization in *Jianfengia* of axonal projections from the anterior sclerite (Fig. 2d) thus identifies the prosocerebral domain of its cerebrum. Conspicuously absent in extant Myriapoda (and in Fuxianhuiidae), the nauplius/ocellar visual system therefore appears to be restricted to certain lineages of Artiopoda[8], Trilobita[9], and Pancrustacea[10,11]. *Jianfengia* further possesses compound eyes that crown each of the paired eyestalks extending laterally from beneath the carapace (Fig. 1b–e). This organization corresponds to eyestalk morphology typifying eumalacostracan crustaceans[15] and other members of Jianfengiidae[16] whose eyestalks evince two articles connected by a suture (Fig. 1c–f). In extant taxa, the combinatorial activity of *Six3, Otx* and *Pax6* defines the protocerebral domain of the pancrustacean forebrain[12,17] and its nested optic neuropils, which receive information from the eye's ommatidial array[18]. Compound eyes on stalks and the areas to which their associated fossilized optic neuropils project thus indicate the protocerebral domain of the jianfengiid cerebrum.

## Reconstructing the brain of *Jianfengia*
Our initial procedure for reconstructing the cerebrum of *Jianfengia* (Fig. 3) followed a routine used previously on the large fuxianhuiid

cerebrum where incomplete neural traces from several specimens were mapped as mirror symmetric profiles within an envelope representing the bilaterally symmetric fuxianhuiid head shield[1]. For *Jianfengia*, the neural traces, manifested as blue-black residues or as clusters of grey to near-black granules, were mapped onto a mirror-symmetric envelope for each of four specimens. The initial envelope was defined by the outline of the right frontal carapace and eyestalk of specimen YKLP11117 (from Fig. 1b). Flipping a copy of the outline across the animal's midline provided the mirror symmetric envelope, as shown in Fig. 3b. The envelope was correspondingly adjusted to map neural traces in specimens that showed evidence of taphonomic misalignments, such as lateral displacements of the eyestalks or medial areas (e.g. specimen YKLP1367: Fig. 3c). These adjustments facilitate mapping of neural traces irrespective of such distortions, as shown for YKLP11368 (Fig. 3d). Likewise, specimen YKLP11369, in which only half the frontal volume could be retrieved, provided neural traces that mapped into the eyestalk and laterally alongside the esophageal foramen (Fig. 3e). To obtain the final reconstruction, the envelopes and their traces of each specimen were readjusted and registered in the summary envelope as shown in Fig. 3g. Finally, each tracing in one half of the envelope was mirrored in the other (Fig. 3h). The symmetrical profile from each of the four specimens was then filled and made 33% opaque. All four profiles were superimposed to provide a summed reconstruction of the *Jianfengia* cerebrum and its connection to the first three contiguous trunk ganglia T1 – T3 (Fig. 3h).

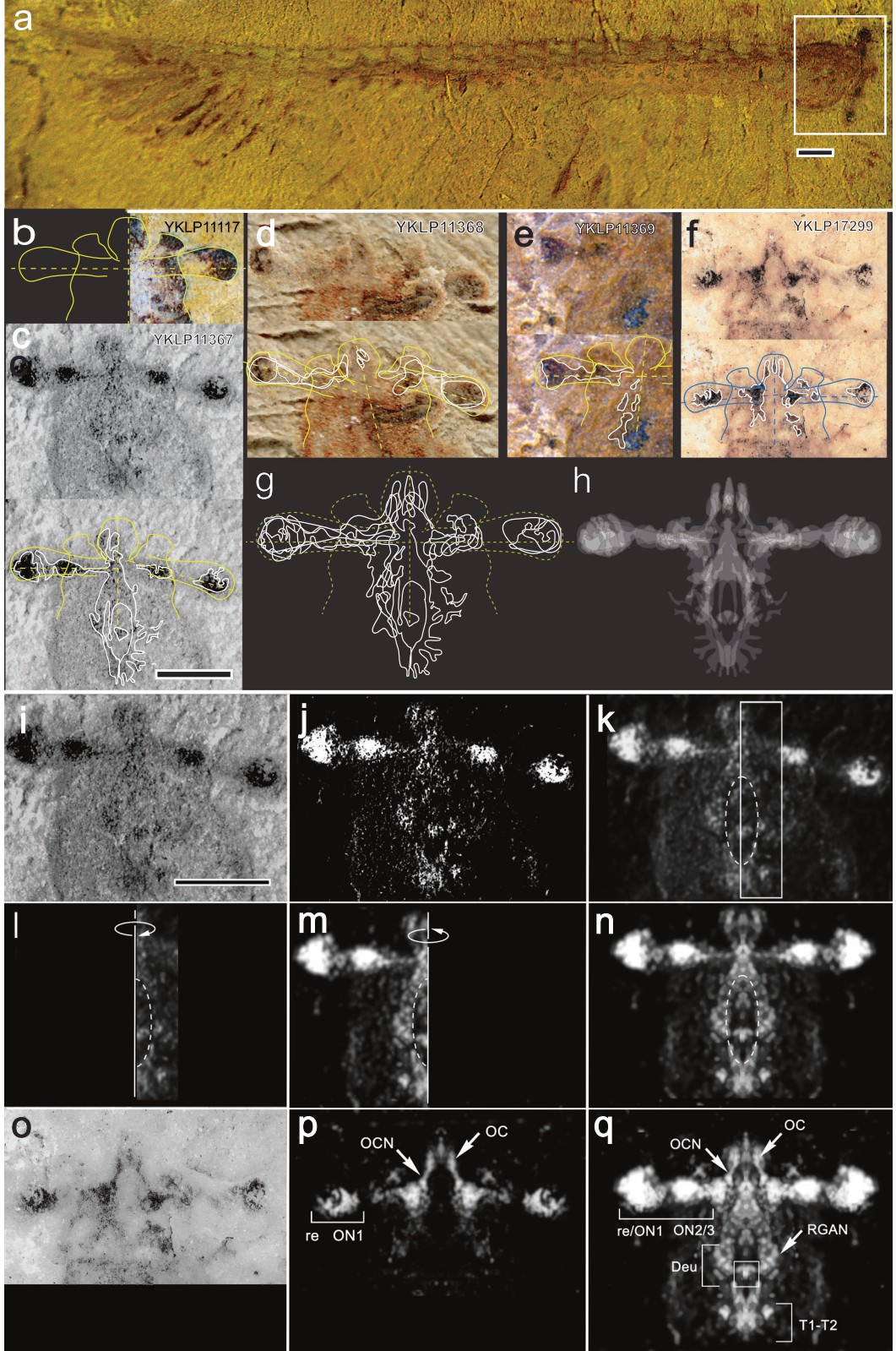

With manual tracing, however, it is impossible to exclude anticipatory bias[19]. To meet this challenge, we generated a second reconstruction using a minimum of human intervention. Two specimens were selected, YKLP 11367 and YKLP17299, both of which provide evidence of bilaterally preserved neuropils and tracts. Unlike specimen YKLP17299, whose mirror-symmetric neural traces suggest it is preserved flat (Fig. 3o), the asymmetric disposition of traces in specimen YKLP 11367

(Fig. 3i) indicates a slight taphonomic rotation around the fossil's antero-posterior axis. Assuming that *Jianfengia* was a member of Bilateria, then these left and right traces in YKLP 11367 each represents a different depth within the specimen and are present on the contralateral side. These neural traces contribute to a mirror-symmetric organization as do the reflected outlines obtained by tracing. Reconstitution required four actions: the elimination of structures beneath a defined gray level (here

**Fig. 3 | Reconstructing the brain of *Jianfengia*.** Optical photographs and tracings in (**b**–**h**). **a** Specimen YKLP11367 showing relative sizes of the trunk and cephalic area (boxed). **b**–**h** Reconstruction based on hand tracings of fossilized neural traces, neuropils, and connections of specimens YKLP11367, 11368, 11369 and 17299. **b** To provide a mirror-symmetrical envelope within which neural tracings were mapped, the outline of the right frontal carapace and eyestalk of YKLP11117 was duplicated and flipped over the midline. **g** All tracings adjusted for axial displacement and mirrored for bilateral symmetry. **h** Tracings from each specimen filled, made partially opaque, and superimposed; the more overlap, the brighter the trace. YKLP11367, 11368, and 11369 provide most of the traces in the eyestalks, flanking the esophageal foramen, and rostral mid-brain. YKLP 17299 provides uninterrupted passage of nerves extending from the anterior sclerite into the prosocerebrum (Fig. 2d). **i**–**q** Reconstruction based on computed intensity levels. **i** Left-right asymmetry of traces reveals partial rotation of YKLP11367 around the antero-posterior (**a**–**p**) axis. Two different levels are exposed at or just beneath the surface of the fossil's fracture plane. Adobe Photoshop set to eliminate intensity levels below the defined threshold of 87% black in standard CMYK scale provides further data after the resultant image was color-inverted (**j**), then subjected to a Gaussian blur function set at $R = 10$px (**k**). To combine the data points from both sides, the right side of the fossil was selected, isolated (**l**) and flipped over the midline and superimposed (**m**) on the left side using the stomodeal outline for alignment. **n** The left-side image is duplicated and flipped over to the right to provide the reconstituted bilaterally symmetrical cephalic nervous system of YKLP11367. **o**, **p** The same procedure applied to YKLP17299. **q**, **p** superimposed onto **n** using the anterior sclerite's ocellar nerves for alignment. The resultant jianfengiid reconstruction (right side shown in Fig. 4a) provides the interpretive diagram shown in Fig. 4b. Scale bar in **a**, **c**, i = 0.5 mm.

87% black of the CMYK scale), thus retaining the darkest puncta; inversion of the image to provide white profiles on a black background followed by the imposition of a Gaussian blur function (here radius expansion $R = 10$px) that expands and runs together puncta (compare Fig. 3j, k). To counteract asymmetry around the midline (Fig. 3k), the right side of the processed image beneath its ipsilateral eyestalk was isolated (Fig. 3l), flipped to the other side and merged with the left half, guided by matching the perimeters of the esophageal foramen (Fig. 3m). The left half was then copied, flipped back to become the right half with the two halves joined at the midline (Fig. 3n). This provides a mirror-symmetric depiction of all traces present in the original specimen (Fig. 3n). Finally, tracings pertaining to the ocellar/naupliar system of specimen YKLP17299 (Fig. 3o) were identically processed (Fig. 3p). The mirror-symmetric image of YKLP17299 was superimposed onto that of YKLP11367, aligning their prominent ocellar features to provide the final reconstituted view of the jianfengiid cerebrum (Fig. 3q). This second procedure precludes subjective bias, yet the resultant image corresponds well with that obtained by tracing.

## Corresponding branchiopod and malacostracan cerebral organization in *Jianfengia*

Both reconstruction methods resolved cerebral neuropils rostral to the esophageal foramen and fossilized nerve cords that extend around the esophageal foramen giving rise to neuropils disposed lateral to it. These nerves converge caudally to provide a synganglion immediately posterior to the foramen. The dispositions of these neuropils as well as neuropils defining the lateral expansion of the protocerebrum into the eyestalks correspond to traits typifying the brains described for the crown groups Branchiopoda and Decapoda[5,18,20–22]. As would be expected from observations of extant eucrustaceans, traces of neural tissue within the jianfengiid esophageal foramen (Fig. 3h, q) also align with the standard locations of the appendicular labrum[15]. In both reconstructions fossilized axon bundles diverge laterally from the anterior sclerite's ocellar/nauplius eyes to supply the rostral neuropil of the prosocerebrum (Fig. 2d) where they merge with bilateral neuropils anterior and lateral to the esophageal foramen. These protocerebral areas receive the eyestalks' nerves which connect a series of three nested neuropils originating from beneath the compound eyes. (Fig. 3h, q).

Next, we determined whether the reconstructed cerebra may correspond to the optic lobes and other neuropils that exist in today's eucrustaceans. Both reconstructions of Fig. 3h, q allow an interpretive view of the jianfengiid brain (Fig. 4a–c). The lateral extension of the brain into the eyestalks comprises a small first optic neuropil (ON1) beneath the compound retina. This center is contiguous with a voluminous second optic neuropil (ON2) nestled against a smaller third neuropil (ON3). That the three neuropils are homologues of the lamina, medulla and lobula typifying pancrustaceans is supported by their alignment and similarity to the eye stalk neuropils exemplified here by the extant decapod malacostracan *Astacus astacus*[23] (upper left in Fig. 4a). The organization of the circumstomodeal nerve cords and their neuropils, however, align with those of branchiopod crustaceans[21,24], such as *Triops longicaudatus* and *Artemia salina*, whose cerebral nervous systems, other than the optic lobes, are nearly identical to that reconstructed for *Jianfengia* (lower left, Fig. 4a). The jianfengiid brain thus features attributes of both the ground pattern of the malacostracan visual system[18] and the deutocerebral-synganglion circumstomodeal nervous system of adult Branchiopoda, which can also be observed during early development of the decapod brain[21,22].

## Is Jianfengia a member of Tetraconata?

A trait claimed to be exclusive to pancrustaceans, and that inspired the term 'Tetraconata' as the alternative name for Pancrustacea[25], is the presence of a quartet of Semper cells in each ommatidium of the compound eye. In living pancrustaceans each Semper cell sends four processes from the base of the ommatidium to secrete the transparent protein that builds the ommatidium's light-focusing cone[26,27]. The presence of such an organization in *Jianfengia* could support its status as a protocrustacean. Specimen YKLP11117 shows a few ommatidia that appear to have lost their cuticular lenses, thus allowing a view beneath them. Combined UV and white light illumination reveal in four ommatidia a geometric arrangement of internal elements that are distinct from possible taphonomic artifacts (Supplementary Fig. 2). Resolving peak intensities in a defined chromatic range (see Methods) reveals grouped iridescent components in four ommatidia. Albeit a small sample, in two ommatidia the groups appear to be organized as a quartet, whereas in the other two ommatidia the resolution is ambiguous in suggesting more than four. Whereas the constrained tetrad arrangement is crucial for the patterning of the pancrustacean compound eye and its ommatidia[26,27], in the Scutigeriidae—centipedes uniquely possessing compound eyes[28]—the Semper cells contribute at least eight prolongations that could provide the crystalline cone[28,29]. The implications that cone cell ambiguities exclude *Jianfengia* from Pancrustacea are considered next.

## Neurocladistics identifies Jianfengiidae basal to total Mandibulata

Arrangements of putative Semper cells do not provide an unambiguous trait supporting pancrustacean affinity. To clarify the phylogenetic status of *Jianfengia* we used neural traits for cladistic determination of its relationship with representative mandibulates and chelicerates. This follows the strategy used for a previous neural cladistics study inferring phylogenetic relationships across extant euarthropods[30]. Here we assembled a matrix of 120 characters comprising mainly neural traits with additions pertaining to the carapace, cephalic appendages, and tagmatization (Supplementary Table 1). The traits were scored as present/absent across 17 living euarthropod species as well as the Chengjiang fossil *Fuxianhuia protensa* and *Alalcomenaeus*[2,7], the Wuliuan stage Kaili fossil *Leanchoilia*[5], the BST fossil *Mollisonia symmetrica*[31] and the fossil limuliid *Euproops danae* from the Carboniferous Mazon Creek Konservat-Lagerstätte[32]. The

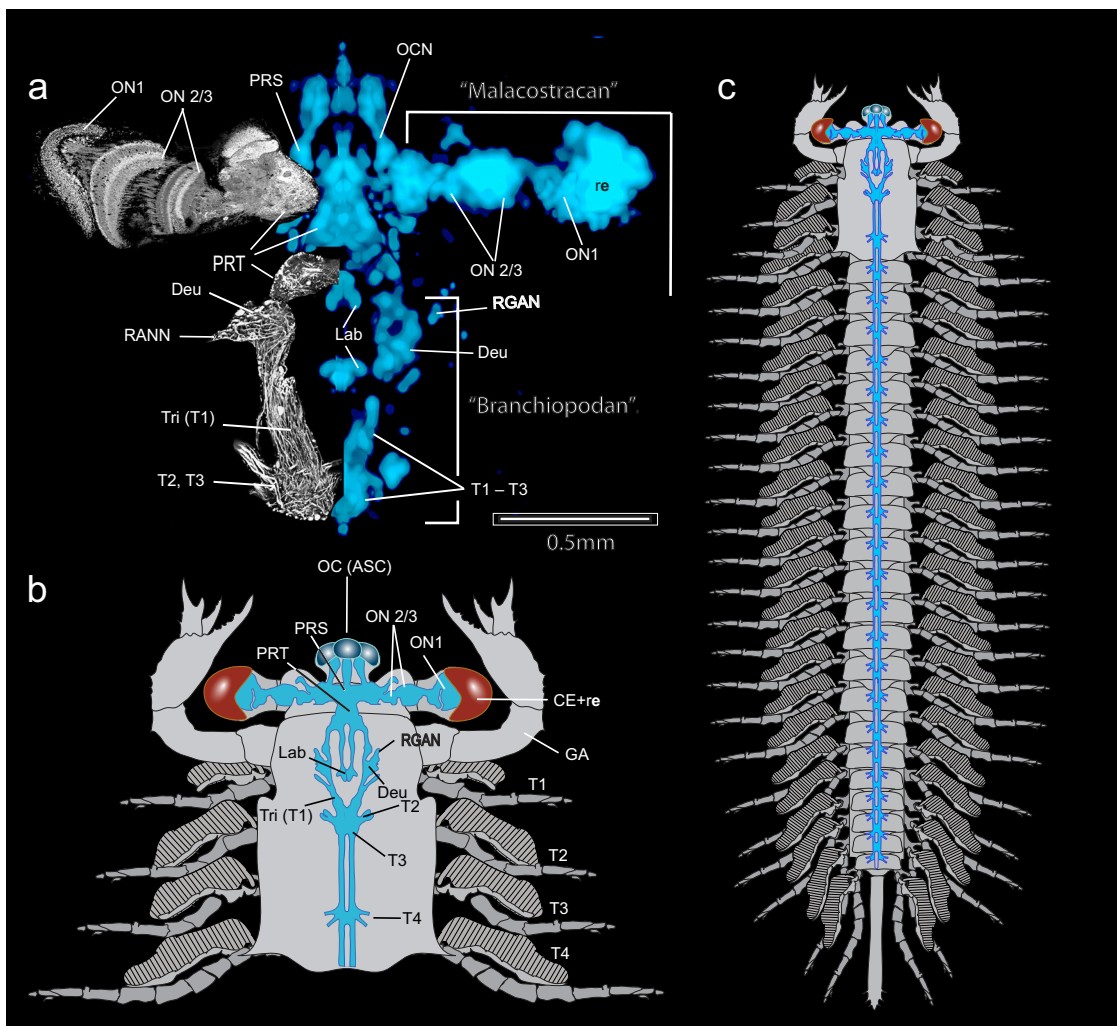

**Fig. 4 | Interpretive reconstruction of *Jianfengia multisegmentalis*. a** Fossil neural traces on the right (blue; scale bar = 0.5 mm) align with arrangements of extant eucrustacean cerebral neuropils (anti-silver-proteinate preparations on left). Optic neuropils of the malacostracan *Astacus astacus* (Rabbit, polyclonal a-Tubulin, Abcam RRID: AB_301787) match the three nested optic neuropils (ON1-ON2/3) and their confluence with the protocerebrum (PRT) of *Jianfengia*. The silver-proteinate section of an extant branchiopod crustacean (*Triops*, lower left) matches the circum-esophageal nerves and the convergence of segments T1-T3 as a synganglion. As in Branchiopoda, the *Jianfengia* deutocerebrum (Deu) is split in half: each half flanks the side of the esophageal foramen. The root of the branchiopod antennular nerve (RANN) corresponds to the root of the *Jianfengia* great appendage nerve (RGAN). The fossil ocellar nerves (OCN) are identified as originating from the ocelli of the anterior sclerite OC (ASC). The paired labral nerves and ganglia (Lab) occupy the same location as in extant pancrustaceans and in Leanchoiliidae, indicating an ancient origin[3,24,55,56]. **b** Annotated interpretive reconstruction of the cephalic area of *J. multisegmentalis*. Abbreviations: Tri tritocerebrum, GA great appendage, OC ocelli, ASC anterior sclerite, ON optic neuropil, CE+re compound eye+retina, PRS prosocerebral domain, PRT protocerebral domain. **c** Idealized reconstruction of whole animal based on specimens YKLP11117, YKLP11367 and YKLP17299. Segmental ganglia have not been resolved caudal to T3. Those shown are hypothetical, based on documented examples in other euarthropods[24]. Trunk appendages are biramous; pairs beneath the carapace have 5-7 articles, the rest have 10, except the last few.

cladistics analysis was rooted against three non-arthropodan out-groups, the spiralian *Paranemertes peregrina*, and the cycloneuralians *Caenorhabditis elegans* and *Priapulus caudatus*. For parsimony and likelihood analyses we employed PAUP* (Phylogenetic Analysis Using Parsimony*, version 4.0a168)[33] to infer evolutionary relationships. To avoid bias, due to the uncertainty of Semper cell numbers that trait was excluded. For maximum parsimony analyses, all traits were unordered and initially considered under equal weighting (Fig. 5). We also performed Bayesian analyses on the same matrix in Mr. Bayes (version 3.2.7a), under the Markov *k* (Mk) model of character evolution[34]. The resultant phylogenetic trees and evolutionary relationships were largely congruent across all inference methods, with minor variation in likelihood and parsimony bootstrap support values and Bayesian posterior probabilities (see Supplementary Fig. 4). Notably, *Jianfengia multisegmentalis* was resolved as basal and sister to Mandibulata in all

phylogenetic analyses with various levels of support (Supplementary Fig. 4), whereas the short body megacheirans *Alalcomenaeus* and *Leanchoilia* were resolved as basal and sister to total Chelicerata. *Fuxianhuia protensa* was resolved as a stem mandibulate closely allied with extant Myriapoda.

## Discussion
Here we have demonstrated that jianfengiid neuromorphology matches the disposition of optic neuropils, neuropils of the proso- and protocerebra, and the circumstomodeal neuropils of extant eucrustaceans (Fig. 4a). We considered whether the malacostracan-like proso- and protocerebra occurring with the branchiopod-like circumstomodeal system of *Jianfengia* might suggest a developmental stage resembling that documented for extant malacostracans[22]. Developmental stages of *Leanchoilia*[35] suggest that at least short-

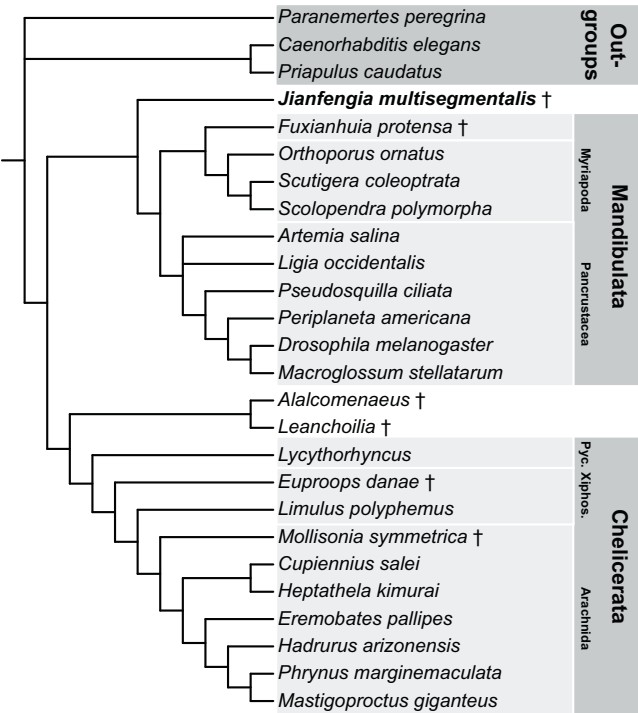

**Fig. 5 | Phylogenetic inferences reveal *Jianfengia multisegmentalis* as sister to total Mandibulata.** (See also Supplementary Fig. 4). Strict consensus tree from maximum parsimony analysis under equal character weights identifies *J. multisegmentalis* as basal sister group and ancestral to all Mandibulata. This tree is the result of two most parsimonious trees (tree length = 221 steps; CI = 0.54; RI = 0.78), with an unresolved polytomy at the base of the pancrustacean clade. Pyc. Pycnogonida, Xiphos. Xiphosura, † = extinct taxon.

bodied megacheirans likely underwent direct development. Specimens of the much rarer long-bodied *Jianfengia* show no size graduation nor any evidence of "Orsten-like" intermediaries that might suggest anamorphic development. In possessing the same number of segments and identical dimensions of the carapace and appendages, the few complete specimens of *Jianfengia* likely represent at least the same stage of development even if not fully mature.

Despite correspondences of the jianfengiid cerebrum and the cerebra of eucrustaceans, there are two confounding aspects of *Jianfengia* that do not align this taxon with Pancrustacea. One is the morphology of the jianfengiid deutocerebral appendages which conform to the 'great appendage' morphologies defining Megacheira[36–38]; the other is that the first two trunk segments of *Jianfengia* lack any evidence of either a second antenna or mandibles. However, in this *Jianfengia* is not alone: leanchoiliids lack specialized appendages belonging to the first trunk appendage (T1, the tritocerebrum) which in extant pancrustaceans provides the second antennae and in chelicerates the pedipalps. Segment T1 is recognized as the anterior limit of Hox gene expression[4] and the restricted expression of the gene *collier*[39]. Although in extant mandibulate arthropods segment T1 may become integrated into the brain because of morphogenetic movements, it is genetically distinct from it and demarcates the interface between the asegmental cerebrum and the segmental trunk ganglia of the ventral nervous system (Supplementary Table 1). The second trunk segment T2 of *Jianfengia* corresponds to the mandibular segment that gives its name to that subphylum. But in *Jianfengia* any morphological trait expected to define a mandible is completely absent whereas in crown Mandibulata their patterning and differentiation depend on region-specific expression of the transcription factor *cap'n'collar (cnc)*, which is restricted by the anterior Hox gene *Deformed (Dfd)*[40–42]. These absences in *Jianfengia*, and the subsequent homonomy of all its post

cephalic trunk appendages, indicate quiescence of early differential Hox gene activity underlying trunk tagmatization[43]. Obviously, however, the arrangement in crown mandibulates of fully differentiated tritocerebral, mandibular, and maxillary appendages must have emerged at some time in early euarthropod evolution. That in *Jianfengia* the tritocerebral ganglion appears to be grouped together with segments T2 and T3 as a synganglion (Fig. 4a) may suggest that differential Hox gene activity was already in play to determine ganglionic differentiation within its central nervous system as it does in the development of branchiopods[22]. The organization of a tritocerebrum contiguous with the mandibular and first maxillary ganglia also marks the jianfengiid nervous system as radically distinct from that of Myriapoda where the intercalary tritocerebral segment in extant groups can be positioned so far forward as to be almost assimilated into the myriapod deutocerebrum[44].

If traits defining its cerebrum suggest *Jianfengia* might be considered as a stem pancrustacean, then why are its deutocerebral appendages not antenniform? The plausibility that an ancestor to Mandibulata could be equipped with stubby great appendages rather than antennules should not be dismissed. Developmental genetics demonstrates that interference with the program defining the formation of aranean chelae—extant homologues of leanchoiliid 'great appendages'—can lead to a genetic reorganization resulting in a switch to a uniramous non-chelate appendage[45,46]. Indeed, both kinds of appendages, chelae and antennules, are uniramous and although most authors emphasize 'great appendages' and antennules as distinct, there are noticeable likenesses between them. One is found in harpacticoid copepods alive today, in which the males develop great appendage-like antennules terminating as two opposing podomeres that function as a clasper to constrain a juvenile female until she is ready for fertilization[47,48].

Any close resemblance of *Jianfengia* to the pancrustacean ground pattern is also unsupported by ommatidial organization. The presence of putative Semper cells that appear to be numerically inconsistent provides ambivalent support for an ancestral arrangement, suggesting that Semper cell development and their number had not yet diversified into the respective character states typifying myriapods and pancrustaceans[25,28]. These ambiguities do not influence the present neurocladistic analysis that identifies *Jianfengia* as sister to total Mandibulata. As shown in Fig. 6, despite differences of their neural and visual system organization the cerebral divisions of Leanchoiliidae and Jianfengiidae align well with each other and with extant Panarthropoda with respect to their conserved order of cerebral domains, associated appendages and their sensory modalities. All cerebral traits are independent of the presence or absence of tagmatized trunk segmentation, which in extant mandibulates is typified by the evolution of three true segments bearing gnathal appendages. In extant Euarthropoda, each cerebral domain is defined by the combinatorial activity of conserved homeobox transcription factors, with *Emx, Nk2* and *Exd* defining the ce3 domain and its deutocerebral integration centers[4], which in Cambrian stem euarthropods equally serve the 'great appendages' or the homologous antennules as in *Fuxianhuia protensa*[7]. Thus, rather than divergences of the segmented trunk or differences in the character state of 'great appendages' it is the invariable order of the asegmental ce1-ce3 domains, their lineage-specific neuropils, and their associated sensory traits that resolve relationships amongst ancestral and crown euarthropods (Fig. 6).

To what degree, then, does a neural cladistic analysis compare with current views of relationships within and external to 'great appendage' Megacheira? To date, it has been features of the cephalic exoskeleton, trunk and appendicular morphologies whose coding has independently led to cladistical analyses that unite *Alalcomenaeus cambricus* and *Leanchoilia superlata* with a variety of other leanchoiliid species to provide the now well-established family Leanchoiliidae[49]. This family is related, but with less support, to the 'great appendage' stem euarthropods *Yohoia*, and *Haikoucaris*[38], both of which have

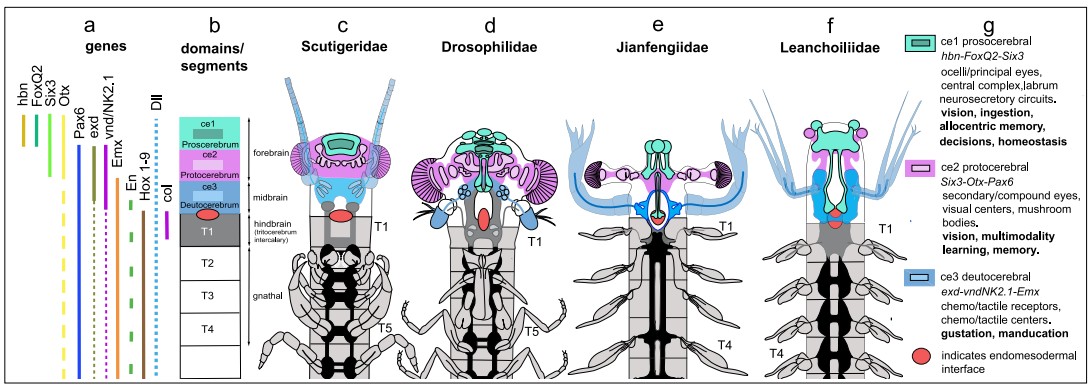

**Fig. 6 | Alignment of extant euarthropod and Cambrian 'great appendage' stem euarthropod brains.** In living euarthropods, combinatorial expression of homologous genes (**a**) defines three asegmental domains of the cerebrum: **b** these are: ce1 (prosocerebrum); ce2 (protocerebrum); and ce3 (deutocerebrum). The first true segment of the trunk is T1, specified by *collier*[39]. All cerebral traits are independent of the presence or absence of tagmatized trunk segmentation, which in extant mandibulates is typified by segments T2-T4 providing gnathal appendages. **c**, **d** Each domain in extant Mandibulata is characterized by its sensory systems and computational centers. In both Scutigeridae and Drosophilidae ce1 is dominated by its central complex. In Drosophilidae and other pancrustaceans[10,11] ce1 is supplied by the rostral visual system (ocelli/naupliar eyes) and provides the paired labra. Domain ce2 supports the protocerebral compound eyes and its nested visual centers and paired mushroom bodies amongst other centers[18,23]. The ce3 domain is served by uniramous antennae and contains chemosensory and mechanosensory neuropils[24]. **e**, **f** Homologous sensory systems supply corresponding domains in two divergent 'great appendage' stem euarthropods. Jianfengiidae (**e**) is equipped with ocelli whose central projections define ce1. As in mandibulates, the Jianfengiid compound eyes and centers define ce2 and the uniramous 'great appendages' define ce3. Leanchoiliidae (**f**) differ in that two pairs of single lens eyes, one forward viewing the other viewing laterally, supply contiguous ce1 and ce2 domains. Paired 'great appendages' define a substantial ce3. **g** Summary of the main sensory and computational attributes of each cerebral domain, as known from studies of extant euarthropods. These are applicable to all investigated mandibulate and non-mandibulate lineages[24]. Alignments of extant and extinct taxa along the shared non-neural endomesodermal interface identifies an invariable order of non-segmented ce1-ce3 domains, their lineage-specific neuropils, and their associated sensory traits, suggesting an ancient organization of domain-specific functionality.

sixteen homonomous segments. Together with Leanchoiliidae these taxa are grouped as the Order Megacheira[49]. That cladistic analysis, which followed an accepted constraint in assigning its considered taxa to Chelicerata[50], also includes the 'great appendage' species *Fortiforceps foliosa*[49] which like *Jianfengia* has at least 27 homonomous segments and stalked compound eyes. It was resolved as a distant sister taxon belonging to the clade Cheliceramorpha which includes Xiphosura and Eurypterida[49,50]. A recent description of the Ordovician leanchoiliid *Lomankus edgecombei* likewise includes *Fortiforceps*, *Jianfengia* and other multisegmented 'great appendage' euarthropods in Megacheira belonging to total group Chelicerata[50].

Those relationships contrast with neuroanatomical traits and neural cladistics that firmly place *Jianfengia* as sister to total Mandibulata, distant from Leanchoiliidae which occupies an equivalent position with total Chelicerata. *Jianfengia*, along with multisegmented 'great appendage' arthropods sharing similar morphologies[51], is removed from any affiliation with Chelicerata. If Jianfengiidae and Leanchoiliidae are otherwise defined by possessing 'great appendages' this is either due to their convergent evolution[52] or because the 'great appendage' generally characterizes lower stem group euarthropods as suggested 23 years ago by a cladistic analysis of traits associated with mouthparts and cephalic appendages which revealed the 'great appendage' morphology as basal to total Euarthropoda[53]. The present findings provide independent evidence for this proposition; they also indicate that 'great appendages' are not exclusively ancestral to chelicerate chela but are also ancestral to the multisegmented uniramous antennules of mandibulates.

## Methods
### Material provenance
Descriptions herein refer to specimens of *Jianfengia multisegmentalis* retrieved from the Cambrian (Series 2, Stage 3) Eoredlichia-Wutingaspis trilobite biozone, Yu'anshan Member, Chiungchussu

Formation, Haikou. The specimens used here YKLP11117, YKLP11367, YKLP 11368, YKLP 11369, YKLP17299 and NIGPAS 100123b are curated at the Yunnan Key Laboratory for Palaeobiology (YKLP), Institute of Paleontology, Yunnan University, Yunnan, Kunming, China. Permission to access and study the material for this work was granted by Yunnan Key Laboratory for Palaeobiology Director.

### Photomicroscopy
For light microscopy of fossil material, digital images were taken using a Nikon D3X attached to a Leica M205C photomicroscope (Leica Microsystems; Wetzlar, Germany). Images were transferred to Adobe Photoshop CS5 (Adobe Systems; San Jose, CA) and processed using the Photoshop camera raw filter plug-in to adjust sharpness, luminance, texture, and clarity. Colors were untouched and are those typical of Chengjiang fossils. For ultraviolet illumination, fossils were photographed using a Leica MZ10 F stereomicroscope with appropriate filter blocks to evoke intense green fluorescence. Flexible fiberglass light guides were used to combine UV fluorescence and white-light illumination. Confocal reconstructions for Fig. 4a were made with an LSM 3 Pascal confocal microscope (Zeiss, Oberkochen, Germany). From 10 to 30 images of 1,024 ×1,024 pixel resolution at 12-bit color depth were scanned by using 10X/0.3 plan Plan-Neofluar objectives. Light microscopy images of Bodian silver-stained neuropils were obtained with a Zeiss Axio Imager Z.2.

### Interpretive drawings and tracings
Drawings were made using Adobe Illustrator for which tracings and reconstruction were generated using a Wacom Intuos Pen Tablet on projected photographic images.

### Identification of putative semper cells
Enlargements of the compound eye of YKLP11117 shows its composition of about 120 facets. Patches of these at the eye margin and about

2/3rds across the eye's surface resolve their hexagonal arrangement. (Fig. 1c, d; Supplementary Fig. 2a, b). Four ommatidia (numbered 1–4 in Supplementary Fig. 2c) show a loss of their capping lenses, thereby revealing structures compressed within the ommatidial shaft. Ultra-violet illumination reveals clusters of bright yellow-green images within four ommatidia. These configurations were then resolved at high magnification by isolating their computed maximum intensities). Between four and possibly seven profiles reside in a sampled omma-tidium allowing their interpretation as arrangements of Semper (cone) cells that would have extended outwards to the crystalline cone (light gray) underlying its lens. Although we schematize quartets of Semper cell as their default organization typifying pancrustaceans, the sample is very small and two of the four ommatidia appear to have more than four units. This indicates the likelihood of variation across the retina.

### Isolating neural traces across specimen YKLP 11367

Imbalances of tone across this specimen were rectified by con-verting the image to its grey-scale mode and removing grey level densities responsible for imbalance (Supplementary Fig. 3). This provided approximate uniformity of grey scales across the speci-men allowing adjustments of exposure and offset to provide a final image for analysis and reconstruction. Adobe Photoshop functions eliminating intensity levels beneath a defined threshold (here 87% black in the standard CMYK scale) provided an image that was grey scale inverted to provide white profiles on a black background. This was subjected to a Gaussian blur function ($R = 10px$) to provide resolution of neural residues free of background noise (Fig. 3n, p). Description of the brain's reconstruction is provided in the test and legend of Fig. 3.

### Phylogenetic inference

Maximum parsimony and likelihood were performed using PAUP* (Phylogenetic Analysis Using Parsimony*, version 4.0a168) on a matrix of 120 characters scored in 26 extinct and extant taxa. Taxa include 17 extant euarthropod species, and 6 extinct euarthropods, including the Chengjiang fossils *Fuxianhuia protensa* and *Alalcomenaeus*, the lean-choiliid fossil *Leanchoilia*, the BST fossil *Mollisonia symmetrica*, and the fossil limuliid *Euproops danae*. Trees were rooted with three extant non-arthropodan taxa. All morphological characters were binary and scored as either present (1) or absent (0). Maximum parsimony was initially conducted with a heuristic search using unordered and unweighted characters employing 1,000 random-addition replicates with Tree-Bisection-Reconnection (TBR) branch swapping. Two most-parsimonious trees (tree length = 221 steps; CI = 0.54; RI = 0.78) pro-vide the strict consensus tree shown in Fig. 5, with bootstrap support values shown in Suppl. Figure 4a. To examine the impact of homoplasy on parsimony trees, subsequent maximum parsimony searches varied character weights by implementing successive reweighting (Supple-mentary Fig. 4b). Successive reweighting was done using the rescaled consistency index until character weighting stabilized, which occurred after three successive rounds of reweighting. This approach increases support for *J. multisegmentalis* as sister to all Mandibulata (Supple-mentary Fig. 4b). Maximum likelihood was performed under the Mkv model, assuming a single substitution type, equal rates, and no invar-iant sites. Likelihood bootstrap analysis was performed on 1000 replicates with support values overlaid on the highest likelihood tree (Supplementary Fig. 4d). Bayesian analysis on the same matrix was performed using Mr. Bayes (version 3.2.7a), implementing the gamma rate variation model with unordered characters. Markov Chain Monte Carlo (MCMC) analysis was run for 100 million generations on four chains with the first 25% of trees removed as a conservative burn-in fraction. Runs converged well with the standard deviation of the split frequencies falling below 0.001. Trees resulting from all phylogenetic analyses were visualized in the Interactive Tree of Life (iTOL) viewer v6[54].

### Reporting summary

Further information on research design is available in the Nature Portfolio Reporting Summary linked to this article.

### Data availability

The specimens used here YKLP11117, YKLP11367, YKLP 11368, YKLP 11369, YKLP17299 and NIGPAS 100123b are curated at the Yunnan Key Laboratory for Palaeobiology (YKLP), Institute of Paleontology, Yun-nan University, Yunnan, Kunming, China. Please contact Professor Xianguang Hou for availability (xghou@ynu.edu.cn). All data reported in this study are included in the article and the supplementary material. Any additional information required to reanalyze the data reported in this article is available from N.J.S. upon reasonable request.

### Code availability

This article does not contain original code.

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

## Acknowledgements

Our special thanks go to Dr. Camilla Strausfeld for her advice and critical editing of the manuscript's penultimate version. Dr. Wulfila Gronenberg gave pivotal advice regarding Adobe Photoshop functions. Dr. Gabriella Wolff prepared and imaged the crayfish optic lobes[23] shown in Fig. 4a. We also thank reviewers for their constructive suggestions. This work was supported by the US National Science Foundation under grant 1754798 awarded to N.J.S. F.H. acknowledges support from the UK Biotechnology and Biological Sciences Research Council (BB/N001230/1).

## Author contributions

X.H. discovered and named the species, provided specimens and with N.J.S. discussed the material. N.J.S. and F.H. originated the project; N.J.S. examined and photographed the fossils. F.H. ascribed published gene expression data to the described panarthropods. N.J.S. prepared the list of characters. D.R.A., with input from N.J.S. and F.H., arranged the data, computed the phylogenetic relationships and interpreted their implications. N.J.S., D.R.A. and F.H. wrote the manuscript.

## Competing interests

The authors declare no competing interests.
