## [Transparent Peer Review file · Nature Communications]

Brain anatomy of the Cambrian fossil *Jianfengia multisegmentalis* informs euarthropod phylogeny

Corresponding Author: Professor Nicholas Strausfeld

Version 0:

Reviewer comments:

Reviewer #1

(Remarks to the Author)
See under review attachments

Reviewer #2

(Remarks to the Author)
This manuscript reports the putative findings of neural traces on a few specimens of a Cambrian great-appendage arthropod: *Jianfengia multisegmentalis*. This species has been described a few decades ago, even though their phylogenetic position within arthropods was not clear. This is not the first time that preservation of the neural tissue in Cambrian arthropod has been used to determine their phylogenetic position. To my knowledge, though, this is the first instance in which a great-appendage arthropods has been clustered together with the groups of crustaceans and hexapods based on these kind of data. Based on this and previous reports, the authors argue that neuroanatomical traits can be used as proxies to understand the placement of these fossils in the absence of external traits, assertion that I agree with.

The authors present a brain and nervous system reconstruction of this species after having superimposed images of neural traces of different specimens and changing illumination parameters. I am not an expert in image treatment but I do not see any apparent introduce software artifact. I find this manuscript as well articulated and with conclusions grounded on the data presented. The images reporting the compound eyes and the neural organisation is consistent with pancrustacean neuroanatomy. I would like to see a bit more of discussion regarding the paraphyly of Megacheira. The phylogenetic placement of *Jianfengia* as an eucrustacean implies that great appendage arthropods can fall either within chelicerates or pancrustaceans. If that is the case, these large appendages were convergent? Lastly, I feel it might be interesting to show in Fig. 3 or 4 stainings of the homologous cephalic area in one extant Eucrustacean species

Lines 57: typo "barely1mm"
Lines 391: typo "Jiangfengia"
Line 445: typo "(C)" should be (D)

Reviewer #3

(Remarks to the Author)
See attached.

Reviewer #4

(Remarks to the Author)
This study analyses four specimens of the species *Jianfengia multisegmentalis* and provides new anatomical information

about them thanks to the use of UV fluorescence and enhanced contrast methodology. This has led the authors to describe three main new characteristics of Jianfengia neuroanatomy, which establish a link to Pancrustacea, thereby improving our knowledge of pancrustacean neuroanatomy and the inner relationships of megacheirans. The three major findings of the paper are:

1. The structural homology of Jianfengia anterior sclerite with crustacean nauplius eyes.
2. The presence of Semper cells in the lateral stalked eyes.
3. The nauplius-like eyes of Jianfengia and its nervous system are connected only in the forebrain region, in a manner similar to pancrustaceans.

The methodology, though I do not consider myself an expert on UV fluorescence, the results, and their interpretation and discussion are solid. UV optical techniques have been frequently used on similar materials to describe the nervous system and, once again, prove to be effective.

Out of the three main results described, I have struggled only with identifying the triplet of lenses in the anterior sclerite (see below). However, due to the more easily identifiable radiated structure of the photoreceptors, which I found clear, I do not question the interpretation of the anterior sclerite as homologous to the nauplius eyes. Therefore, I concur with the three major findings of this work.

I also believe that the contribution of this study to our knowledge of early euarthropod neuroanatomy, and its role in phylogenetic assessment, has the potential to significantly aid in resolving the relationships of major clades and in clarifying megacheiran classification. This contribution is more than significant and is likely to be useful for further studies. As such, I recommend its publication in Nature Communications.

Nevertheless, I suggest a minor revision to improve the exposition of the results and to enhance the discussion in some areas.

-Lines 33-35: Add citations for Myriapoda, the oldest evidence of neuroanatomy.

-Lines 51-52: I would smooth this phrase and say something like, "Together, these findings suggest Jianfengia as a stem euarthropod closer to pancrustaceans than to both Fuxianhuia and Alalcomenaeus." Again, it is not that I am not convinced of the results, but rather that there is conflicting evidence in other characters.

-Line 57: "barely1mm" should be "barely 1 mm."

-Lines 99-102: Rephrase and possibly split the sentence for better readability.

-Line 391: Missing caption for le.

-Figure 5: It would benefit from the addition of a schematic of Fuxianhuia. Also, consider using a hand-drawn cladogram based on your findings and the literature to illustrate the possible evolution of the anatomies depicted.

-Figure S1: Please provide better contrast for the triplet of lenses, at least in the drawing. The structure is difficult to discern here and in Figure 1. Maybe add schematics to help the reader visualize what you see.

-I strongly encourage the author to add a paragraph in the discussion about the possible implications of ontogeny for the anatomy of the nervous system and eyes in Jianfengia. Considering the possible life stage of the studied specimens, as well as of the other key taxa, could provide a clearer understanding of the evolutionary forces at play. Since many characters change during ontogeny, we do not always know whether the specimens represent fully developed adults, and character development may occur with shifts in relative timing among different groups. Addressing this aspect could help reconcile some of the conflicting signals observed in different traits (e.g., antennae/great appendage expression) and offer a clearer perspective on the mosaic evolution seen among Fuxianhuia, Alalcomenaeus, and Jianfengia itself.

I do not wish to see the manuscript again, thus I will be happy to do so if the editors will consider it

relevant. Lorenzo Lustri

Version 1:

Reviewer comments:

Reviewer #1

(Remarks to the Author)

This has become an excellent paper now. With great respect, we note that following our suggestions, this paper has been rewritten and has been changed in crucial points, as assigning Jianfengia basal, but not within the Mandibulata, because the cone-cell number had not yet evolved within Jianggegiidae. Now Jianfengia's position within Euarthropoda is established through a solid, diligent, and careful cladistic analysis, and in particular neuroanatomical and cephalic traits. The results are absolutely convincing and well exposed. Perhaps one could have more emphasized that Jianfengia obviously has a transitional state between stem and crown-group arthropod, because it bears features of both (number of cone cells vs, brain system).

Great work – my congratulation.

I found some small points:

Line 32 end: a "t" too much

Line 92, 93: This organization corresponds to eyestalk morphology typifying eumalacostracan crustaceans and ...

We will find out later (l. 152 ff) why this is mentioned here, but at this point it may seem a bit suggestive. I would suggest to mention that these stalked eyes, as typical of other Cambrian arthropods, also, correspond to those of modern eumalacostracan crustaceans ...

Line 153: Both -> both

Reviewer #2

(Remarks to the Author)

The authors addressed successfully all questions and concerns of the four reviewers. I already liked the previous version, but after applying the suggestions by different experts I consider this manuscript as more robust and I would like to see it published in its current form.

Reviewer #3

(Remarks to the Author)

Reply to Reviewer #1

PD Dr. Brigitte Schoenemann

A pancrustacean brain and its visual systems define a Cambrian “great appendage” arthropod.

Nicholas J. Strausfeld, Xianguang Hou, and Frank Hirth

As its predecessors, this manuscript succeeds excellently in revealing fossilized nervous systems, sometimes, as here, in very small organisms and more than half a billion years old. The authors aim to show that coevally lived stem and crown arthropods during the Early Cambrian in the Chengjiang Biota by the example of *Jianfengia multisegmentalis* (Hou, 1987), a not (completely?) tagmatized arthropod. Two more characteristics should be fulfilled to assign *J. multisegmentalis* to pancrustaceans— a tetraconate crystalline cone and a pancrustacean brain. We agree with the nervous system's interpretation but consider the discussion of the eye systems as problematic. Finally, we come to a conclusion, different from the authors, as stated below:

We thank the reviewer for her positive appraisal of this work and much appreciate that she chose to be visible to the authors. That a reviewer doesn't feel the need to hide behind the shield of anonymity makes discussions such as the present reply informative and relaxed.

We also agree with this reviewer that the section on cone cells needed serious reconsideration, which we have done here to the advantage of the work as a whole. Agreeing with the reviewer regarding ambiguities of the number of cone cells has led to our major revision which is a cladistic analysis of *Jianfengia*'s position, hence identity, within Euarthropoda using neuroanatomical and cephalic traits specifically *omitting* cone cell numbers for the presented phylogenetic analyses. This results in a phylogeny that positions *Jianfengia* basal to, but not within, Mandibulata. It follows that taxon-specific determination of cone cell number had not yet evolved within *Jianfengiidae*. This is now explained and discussed in this revision.

This is an ambitious manuscript that aims to show that Early Cambrian fossils from the Chengjiang Biota reveal recognizable crown groups, namely representatives of tetracornates/panarthropods, already existing coevally with early stem arthropods (e.g. *Fuxianhuia protensa* Hou, 1987 [stem mandibulate], *Alalcomenauis* [stem chelicerate]) more than half a billion years ago.

The authors describe 3 specimens of the genus *Jianfengia multisegmentalis* (Hou, 1987). To investigate neural traces indicating shapes of brains, ganglia, and neural pathways within a carapace barely 1mm wide may be considered quite a challenge. Despite the

absence of trunk tagmatization, the latter would be typical for pancrustacea/ tetraconates; in addition, the authors aim to demonstrate typical traits of pancrustaceans in *J. multisegmentalis* to provide a 'stemward' identity towards Pancrustacea/Tetraconata as there are: Structured eye stalks (-> crown Malacostraca): a) Compound eyes with ommatidia, revealing tetradic structures (-> cone- building Semper cells); b) nauplius-like eyes and associated nerves supplying a discrete forebrain region-> pancrustaceans; c) neuropils and tracts in the eye stalks -> nested optic neuropils as occur in eucrustaceans organization of the deuto- and tritocerebrum -> differentiation from stem Chelicerata and extinct Myriapoda. If successful, the authors would show that neuroanatomical traits could provide a powerful tool for distinguishing characteristics to establish euarthropod relationships, even if external traits that differentiate identities are absent.

The reviewer is correct in identifying the aims of this work, with this revision going significantly further. The over-arching question we addressed, and in this revision have resolved, is whether a brain whose character traits are essentially eucrustacean but belonging to a group that lacks mandibles can be counted as indicating an upper stem eucrustacean or a basal taxon allied to Mandibulata. It also addresses whether an euarthropod *not* equipped with antennules but with chelicerate 'great appendages' could be considered allied to Mandibulata (we show in the revision that it can, as can a lack of mandibles).

Thus, the question what is Jianfengia? To resolve this, we undertook a major revision of the original study: an exploration of the phylogenetic relationship of Jianfengia within the context of Chelicerata and Mandibulata. This new phylogenetic section of the manuscript demonstrates that *Jianfengia* is indeed sister (thus basal) to total Mandibulata. The other "Great Appendage" clade, Leancoiliidae, is basal and thus sister to Chelicerata. The explanatory power of neurocladistics reaching this result further emphasizes that consideration of cone cell numbers was not necessary to place Jianfengia where it is now is.

Genetic studies, which we cited in the original manuscript and again here, indicate that the genetic circuit underling the development of chelicera requires few substitutions to switch the cheliceral phenotype to a uniramous non-cheliceral phenotype. Thus, minor genetic innovations would have provided a subsequent jianfengioid-like ancestor with bone fide antennules. But the most interesting finding resolved by the now added neural cladistics is that it demonstrates that Jianfengiidae being basal to Mandibulata means its 'great appendage' must have preceded the evolution of mandibulate antennules. (Likewise, Leancoiliidae being basal to Chelicerata means its 'great appendages' were ancestral to the arachnid chela).

This is a difficult review because, of course, the manuscript presents its own interpretation of the anatomical findings. Still, another view in the literature sheds a slightly different light on some aspects. In good academic tradition, I will not attack this manuscript but briefly explain this different perspective. Based on this, I will discuss the results presented in the manuscript, and ultimately, the reader may take his or her position

This is rather ambivalent comment from someone who most definitely makes the point of thoroughly interpreting her own anatomical findings of Trilobite visual systems. We assume that Dr. Schoenemann shares with us the assumption (and hope) that readers

expect descriptions of anatomy to include the authors' interpretation of what is significance. Without authorial interpretations of empirical data all our work would be worth far less.

So, what has been achieved?

a) Structured eye stalks (-> crown Malacostraca)

The joints of the articulated eye stalks can be seen very clearly. The authors indicate one, and there is another more distally (Fig. 1F). There are mentioned 2 (line 78), and an arrow should also indicate the second one.

The other is of course the margin of the compound eye itself with the eyestalk cuticle but I am not sure if this is a true suture indicating the eye itself is derived from a modified article.

It is commonly accepted that Malacostraca possess stalked eyes, and at least among the extinct and fossilized I know, the stalks are indeed always articulated. The other way around, I would expect that among the early arthropods, the artiopods, for example, we may find articulated eye stalks. So, I would be slightly more cautious with the conclusion drawn here [Meanwhile, these articulated stalked eyes have been found and described: Schmidt, Schoenemann, et al. 2025, *Biology Communications*, in press]. So probably articulated eye-stalks are typical for crustaceans (not hexapods, belonging to Pancrustacea also!), and consequently, not all arthropods with articulated eye stalks are pancrustaceans. Even the eye stalks of *Leanchoilia* and *Alalcomenaeus* are articulated, see Fig 1 C,D.

We have enjoyed the Schmidt. et al paper. However, there is possibly a misunderstanding. We don't assert *anywhere* that arthropods with eyestalks *must* be indicative of pancrustaceans. Consider isopods, for example. Our own studies of *Leanchoilia* and *Alalcomenaeus* identified pendulous eyestalks as did previous studies by others (e.g: doi.org/10.1016/j.asd.2015.07.005). Whether they were articulated, and thus had the ability to move (scan) in life is wholly unresolved for any fossil as it is for *Jianfengia*. We make no claims that any ancient eyestalk was articulated *sensu* Stomatopoda (which is the only substantiated example of individual eyestalk motility in any arthropod, living or dead). Of phylogenetic interest is that the presence of sutures along the length of a pronounced eyestalk suggests *derivation from* an articulated ancestry. That's not our original idea but was suggested much earlier, particularly by R.E. Snodgrass (Comparative Studies on the Head of Mandibulate Arthropods. Comstock, Ithaca, NY. (1951). We do not contest Dr. Schoenemann's observations of eyestalks in other fossil species. We are also sure that Dr. Schoenemann would agree that fossils cannot provide convincing evidence for motility. Afterall, data about eye stalks are from animals that were likely dead or dying, covered with fine material, and then petrified in time. Who knows whether the moribund attitude of the eyes displaced their angular disposition vis-à-vis the carapace. Trilobites of course don't present that problem, fortunately.

Figure 1 Articulated eye stalks of the artiopod *Pygmaclypeatus daziensis* Zhang, Han & Shu

2000, of *Leancoilia*, and *Alalcomenaeus*,

Yes, these are nice examples. In every case the postmortem attitude is that of a pendular attachment, which as Dr. Schoenmann has noted are distinct from laterally extended eyestalks typifying Eucrustacea. All we are saying is that *Jianfengia*'s eyestalks conform to a eumalacostracan type disposition. We don't see any conflict with the paper cited above or with Dr. Schoenmann's own work or others she cites.

a) Compound eyes with ommatidia, revealing tetradic structures (-> cone-building Semper cells)

The term semper cells defines cone cells of crustaceans and insects. The manuscript develops the idea of including *J. multisegmentalis* within the stem Pancrustacea. The manuscript should take into account the initial and provisional nature of the use of this term.

The question of the existence of tetradically arranged cone-cells in *J. multisegmentalis* is an important, critical point. Indeed, *J. multisegmentalis* possesses compound eyes, and the loss of cuticular lenses in specimen YKLP 11117 offers insights into the arrangement of internal structures of the ommatidia. Photoprocessing methods were applied to clarify the individual elements. However, if I were to take the position of devil's advocate, I could say that the threshold can be adjusted until 'it fits' (I am sure the authors haven't done this.

We explain in the suppl. methods that obtaining the locations of maximum intensities in a color image requires not selecting a general "threshold" but choosing a narrow band of the image's color spectrum. It's a matter of identifying that part of the color range that is distributed across the sample and hence serves not to discriminate but to find inclusiveness in providing the empirical data set. It is not an "until it fits" procedure!

It is phenomenal that the authors succeed here in showing the cone cells. The problem for me here is that none of the examples clearly shows 4 cone cells; one may find even more at different sizes and numbers (Fig. 2). That the application of photo-processing brings out the tetraconate pattern for me suggests, in conclusion, that there are more than 4 (~6-8) cone-cells, but some of them, four in number, are more pronounced in their development than the others. This may indicate that we here are on a transitional way from a multiconate eye (-> *Limulus*-like (Xiphosura, Chelicerata), ~ 100 cone cells, Fahrenbach 1968, Harzsch & Hafner 2006) to a tetraconate system. But still, it is not achieved yet.

We agree entirely (see below)

In the manuscript, the system here is distinguished from the scutigrid myriapods by mentioning one single extension of the cone cell contributing to the secretion of the crystalline cone in tetraconates, while the myriapod possesses two. I am pretty sure one may not find these delicate structures in a fossil. Here, it may help with the argument that the scutigrid ommatidium is developed secondarily from the myriapod ocelli, which in turn originated from original ommatidia (e.g. Paulus 1979, 2000).

Why shouldn't they be visible in a fossil? Dr. Schoenemann proposes that there are more than 4 such prolongations in our own data which surely obviates skepticism.

Müller et al. (2003) find for the myriapod *Scutigera coleoptrata* (Linnaeus, 1758), 8 cone segments, produced by four crystalline cone cells; tripartite cones (Ascothoracida and Cirripedia), and bipartite ones they suggested to have derived from the tetrapartite system. Thus, four cone cells alone may not be sufficient to postulate an assignment to Tetraconata, even if it is the standard definition. At least this should be discussed more differentiated. It may well be that not all cone cells/segments can be found here in the specimens of *J. multisegmentalis*; in one of the examples presented in the manuscript (Fig. 2,4), there may be evident even 8 elements (D4, another shows up 7, D3), suggesting a tetrapartite origin as in *Scutigera*.

We appreciate Dr. Schoenemann's expertise regarding the broader presence of cone cells, which we agree must have very deep origins, preceding divergences of Myriapoda and Pancrustacea. That is also stated clearly in the original manuscript and in this revision. Dr. Schoenemann's commentary above reflects her own scholarship, and we appreciate her request for caution in assigning the status of "crustacean" or "myriapod" cone cells. As already mentioned in this reply, for this revision we have down-played cone cells as reliable indicators. Instead, this has promoted our neurocladistic analysis based on neural or allied traits to determine *Jianfengia*'s phylogenetic position. Notably, to exclude bias *we omit traits suggesting numbers or restrictions of cone cells*. In the same spirit of caution, we have relegated the figure showing possible cone cells to the supplementary figures. The neurophylogenetic analysis reveals *Jianfengia* as sister to total Mandibulata, which includes Tetraconata and Myriapoda. This excludes any claim that *Jianfengia* might be a Tetraconate.

b) nauplius-like eyes and associated nerves supplying a discrete forebrain region -
> pancrustaceans

It would be great to have them, and I am willing to find them in the figures of the main text; I have tried hard ... I see the filamentous structures, and they may be relicts of a palisade of receptors, and I agree with the pigment. Still, please make the 3 nauplius-like eye elements more evident in your main text figures. I find them in the UV (S1 C). A probable explanation indeed would be that this was one organ consisting of three ocellic covered by a membrane, which would explain the homogenous appearance of the whole system under white light. I agree with you that the organ may have been movable. The movable character, I think, results from the not-fixed 'stalk,' which clearly protrudes from the carapace. See the dark line (the margin of the carapace, indicated by the arrows in our Fig 2c).

We are very glad that in view of her own expertise Dr. Schonemann agrees with our identification of ocelli and ocellar nerves. We have modified the UV image accordingly to show the most plausible fit of three ocelli and this figure now appears where the reviewer suggests, namely as Fig. 2.

The small element in the reconstruction drawing (main text in Figure 4D) shows the same interpretation as I gave it above and in Fig. C3, which is great, of course, but this

finding should be established more clearly in the main text.

If we accept that the filaments are relicts of receptors, please note that they do not show any grouping. Typically, a Nauplius eye is a single median eye and consists of three- or four-pigment cup ocelli. This nicely can be detected in your Figure S1C. I am not sure about the arthroal membrane here, but I share the idea that this whole structure could have been movable (see above).

I think these results (3 ocelli, movable system) should be more emphasized and shifted from the Supplement to the main text.

So, we agree with a tripartite, nauplius-like eye, probably built by ocelli.

We agree that the evidence for the nauplius eyes is convincing and have now placed this as Fig. 2 in the main narrative of the work. We thank the reviewer for the suggestion to do so and are glad we concur on this important finding.

Neural organization

c) neuropils and tracts in the eye stalks -> nested optic neuropils as occur in eucarustaceans

d) organization of the deuto- and tritocerebrum -> differentiation from stem Chelicerata and extinct Myriapoda

Tracing up the neural structures in this tiny organism, neuropils as pathways is fascinating, very skillful, and convincing. One should never forget that all these investigations start in a quarry, and we deal with 'rocks', fossils. The optic neuropils 1-3 can be discerned very clearly, the reconstruction of the entire brain for example by overlaying the results of different findings and the assumption of a bilateral organism, by that, the methods of reconstruction for completing, for example, by flipping elements of the one side to the other, are justified. The tables (S1, S2) are interesting, and the drawn reconstructions (Fig 5) are excellent and consequent. Unfortunately, the inscriptions in Figure 5 (as perhaps the whole figure) are too small and hardly readable/discernible.

We are glad that an expert on fossilized visual systems finds our identification of neural traces convincing. The revised work now has an expanded analysis of these microscopic neural traces, using four specimens rather than just two. We have employed two strategies for reconstructions: one using human tracing “by eye”; the other using Gaussian amplification of traces comprising dark deposits. It is gratifying that the two strategies converge. These two sets of results are now shown in Figure 3. We also include a supplementary figure (Supplementary Fig. S1) illustrating how Gaussian blur can be utilized in revealing discrete structures.

I am not an expert in arthropod brains and the genetic background of the genetic organization of an arthropod segmentation, especially the nervous system, but one can follow the given reasoning, and a possible interpretation seems to have emerged.

That the character identity networks (ChINs) are shared across total Euarthropods suggests a pre-Cambrian origin of the genetic ground pattern of the cerebrum. This will

become clearer as more fossilized brains are discovered and which show evidence of structures specific to one or more of the genetic networks known to be shared across this phylum

In Figure 3 it is not clear what is from YKLP 11367 and YKLP 17299

Thank you for pointing this out. We have modified the text to clarify this.

Discussion

The discussion of the manuscript makes the point that the *J. multisegmentalis* among

the stem-arthropods of the Chengjiang biota represents a true pancrustacean, because, although a trunk tagmatization still is missing, the eyes have a tetraconate ommatidium, *J. multisegmentalis* possesses as typical for pancrustaceans a nauplius-like eye consisting of three elements, and a brain that is organized as in pancrustaceans.

We may agree with the latter, but we see a problem with the interpretation of the eye systems.

Note that median eyes (-> Nauplius-like eyes, Nauplius eyes = Median eyes of crustaceans) also exist in the megacheiran *Leancoilia*! (Garcia-Bellido & Collins 2007, Schoenemann & Clarkson 2012, 2023), and indeed in *Alalcomanaeus* (see below).

We may have been too obscure in the original discussion. Our present revision makes it clear that when it comes to sorting out phylogenies, two visual systems are recognized and coded as two separate systems across Chelicerata and Mandibulata. The median eye pairs of course exist in Megacheirans. And we note that Dr. Schoenemann has discovered the same rule for Trilobites, thus aligning as an established fact that this arrangement occurs across Euarthropoda with the crucial exception of Myriapoda. Along with Fuxianhuia, myriapods have no rostral eyes. There was no genotypic data available for Paulus when he was debating such matters. The absence of ocelli in myriapods appears to be a genuine evolved loss. We refer to the relevant 2015 paper that further describes those two sets of eyes: the frontal pair belonging to the anterior part of the forebrain, the prosocerebrum (as do pancrustacean ocelli and nauplius eyes). The lateral pair (secondary eyes in short body megacheirans) belongs to the posterior part of the forebrain, the protocerebrum. The presence of two genetically distinct visual systems, one prosocerebral, the other protocerebral, has been widely recognized, including by Dr. Schoenemann and her co-author regarding the identification of frontal ocellar-like eyes and lateral compound eyes in Trilobites. It's an important work confirming that two different visual systems must very ancient. A number of studies of Radiodonta further suggest the immense age of these two systems. We hope that our modification of the revised manuscripts clarifies any ambiguities suggested by the original submission.

Figure 4 Median eyes of *Alalcomanaeus cambricus* Simonetta, 1970 and *Leancoilia superlata* Walcott 1912

This has implications and should be considered and discussed.

See above responses

Even being aware of the work of Tanaka *et al.* 2013, one may be a bit careful with the interpretation of the eyes of *Alalcomenaeus* and *Leanchoilia*. The specimens of *Alalcomenaeus* illustrated by Briggs & Collins (Fig 4.1, 4.4) or of Whittington (1981, Fig. 126) clearly show that *Alalcomenaeus* has club-shaped, penduculate compound eyes (here Fig. 1D), surmounting articulated stalks; the same is true for *Leanchoilia* (here Figure 1 B, C). Here, the eyes and their stalks are much finer.

Note that there are median eyes in the megacheiran *Leanchoilia* (e.g. Garcia- Bellido & Collins, 2007; Schoenemann Clarkson 2012) and in *Alalcomenaeus* also (e.g. Briggs & Collins, 2007, Fig. 4.3), as can be seen clearly here in Figure 4. This has implications and should be considered and discussed - for example, consequently, *Jianfengia* is NOT the only Early Cambrian genus identified with median eyes (and crustacean-like eyestalks). Perhaps *J. multisegmentalis* is among the first with 3 median eyes. Still, there are crustaceans with 4 ocelli in their median eye complex, e.g. among the branchiopoda (e.g., Reimann & Richter 2007) – so the number alone may is not game-changing here.

Regarding these two paragraphs: we are not sure why Dr. Schoenemann is emphasizing these very clear distinctions. They are well known to many working on arthropod vision including ourselves. So, we do not know where or what in our manuscript would elicit those remarks. Our descriptions of the megacheiran visual system fully accords with Dr.Schoenemann's views and the papers she cites. We have published elsewhere that *Jianfengia*'s visual system is described as one example common to Jianfengiidae, which includes *Fortiforceps foliosa* and *Pseudoiulia (Sklerolibyon maomima)*. Again, *nowhere* are we claiming that *Jianfengia* is the only Early Cambrian genus identified with median eyes and crustacean-like eyestalks. That is explicit in the present work. It is, however, crucial to distinguish eyes on robust eyestalks, which is a trait of all three species of Jianfengiids. Further, we took care to describe the range of ocelli that occur across pancrustaceans, bearing in mind that in insects, and probably in eucrustaceans, two pairs of embryonic precursors each side of the midline provide either 4 ocelli, or multiples of four as in some reptantians, or three ocelli as in Hexapoda, the middle ocellus being a developmental fusion of the two embryonic ocelli either side of the midline. Thus, we see no differences of opinion or conclusions between our description and the works cited by Dr. Schoenemann or her own studies. The same goes for *Leanchoilia* and *Alalcomenaeus* except theirs' are not strictly "median eyes" but are prosocerebral principal eyes, both of those genera being sister to Chelicerata.

We agree, however, completely with the idea that genetically based the morphological variance of homolog deutocerebral appendages is wide enough to comprise great appendages, antennules formed as 'pinchers' or pier-like antennules, or ...

In general, we also agree with the interpretations of the neural systems and the homologies explained here. I am not an expert in arthropod nervous systems, but I think the deductions are reasonable.

We are glad that the reviewer agrees. We suggest viewing Figure 4 of the revised manuscript. It reveals striking correspondences of the jianfengiid cerebrum with the optic lobes and circum-stomodaeal pathway of both malacostracan and brachiopod arrangements.

Finally, one of this manuscript's great conclusions is that not the Hox-gene-associated

divergences of the segmented trunk define ancestral and crown arthropods, but do the segmental cerebral domains, the unique lineage-specific neuropils, and associated traits. This is an important point, but our conclusion differs slightly from what the authors indicate here.

It is a pleasure to read that comment. It is one made by a biologist rather than a paleontologist who might not be so conditioned with regard to gene expression studies that distinguish genetic determination of the trunk versus the cerebrum. We are not claiming the cerebrum is segmented. Snodgrass clearly rejected that in his embryological studies where he observed no true segments in the cephalon, his nomenclature for the cerebrum. Heymons, much earlier in 1901, published the same observation from studying embryonic development of *Scolopendra*.

Even if one of the characteristics (like those of the nervous system as here) has passed the 'finish-line' to reach a crown group state, while others have not (tagmatization of the body), one has to be aware that as here, obviously boundaries of the stem-/crown groups cannot always be rigidly defined, and that there are transitional situations (as represented in the eye-status of *Jianfengia multisegmentalis* (Hou 1987)). Here, one part (brain system) has reached that stage, and the other (tagmatization of the body) still has not; the eye system of *Jianfengia multisegmentalis* (Hou 1987) is on its way. This may have important implications for describing evolutionary processes in general.

Exactly. The cardinal observation. Thank you.

The results here show that the formation of different crown-group characteristics (tagmatization of the trunk – crown–group–nervous–system – eye system) may evolve with different velocities. An important point because one now will have to weigh when the complete, or a 'sufficient' crown-group-state has been arrived. Probably it is not adequate if only one of the characteristics has passed the 'finish line'. Even if the focal point in the considerations of this manuscript has been laid on the eyes and brain/nervous structures, it may be an essential fact that the tagmatization of the body is not quite there yet. Nevertheless, the authors are right to state that neuroanatomical traits could provide a powerful tool for distinguishing characteristics to establish euarthropod relationships, even if external traits that differentiate identities are absent.

Indeed, all these considerations are important. And they open interesting avenues down the road. Hence the utility of providing neural phylogenetics that implies sequential branches of gradual change towards crown morphologies.

Reply to Reviewer #2

Reviewer #2 (Remarks to the Author):

This manuscript reports the putative findings of neural traces on a few specimens of a Cambrian great-appendage arthropod: *Jianfengia multisegmentalis*. This species has been described a few decades ago, even though their phylogenetic position within arthropods was not clear. This is not the first time that preservation of the neural tissue in Cambrian arthropod has been used to determine their phylogenetic position. To my knowledge, though, this is the first instance in which a great-appendage arthropods has been clustered together with the groups of crustaceans and hexapods based on these kind of data. Based on this and previous reports, the authors argue that neuroanatomical traits can be used as proxies to understand the placement of these fossils in the absence of external traits, assertion that I agree with.

We are very glad that the reviewer endorses that strategy.

The authors present a brain and nervous system reconstruction of this species after having superimposed images of neural traces of different specimens and changing illumination parameters. I am not an expert in image treatment but I do not see any apparent introduce software artifact. I find this manuscript as well articulated and with conclusions grounded on the data presented. The images reporting the compound eyes and the neural organisation is consistent with pancrustacean neuroanatomy. I would like to see a bit more of discussion regarding the paraphyly of Megacheira. The phylogenetic placement of *Jianfengia* as an eucrustacean implies that great appendage arthropods can fall either within chelicerates or pancrustaceans. If that is the case, these large appendages were convergent? Lastly, I feel it might be interesting to show in Fig. 3 or 4 stainings of the homologous cephalic area in one extant Eucrustacean species.

These comments are much appreciated as is the suggestion that we provide homologous cephalic arrangement in extant eucrustaceans. This we have done, and it is now shown in Fig. 4a, and discussed in the main narrative. Regarding the distinction of Megacheira: accepting the metric distinctions of the short bodied Leanchoiliidae and the multisegmented *Jianfengiidae*, our identification of neural traits in the latter allows us to use these and those already documented for Leanchoiliidae to explore *Jianfengia*'s phyletic position within the euarthropod tree. Our neurophylogenetic analysis comprises one of the major revisions of this study as it allows us to put to one side the somewhat ambiguous implications of the presence of cone cells. Those ambiguities, raised by one of the reviewers, required a phyletic reassessment in ascribing (or rejecting) a pancrustacean-like identity to *Jianfengia*. Employing neural cladistics while omitting cone cell numbers as an unreliable trait provides a crucial finding. Namely, cladistics places *Jianfengia* as sister to total Mandibulata whereas Leanchoilia and *Alalcomenaeus* together representing short-bodied

megacheirans that occur as sister to total Chelicerata. This addition to the present study adds an important and possible novel observation regarding the evolutionary relationship between Jianfengiidae and Leanchoiliidae within the overall organization of the two major branches of the euarthropod tree of life. It also raises the interesting possibility that "Megacheira" should no longer be considered a clade but instead the two types of 'great appendages' as convergent character traits of deutocerebral appendages. We address this possibility in the revised discussion of the manuscript.

Lines 57: typo "barely1mm

OK

Lines 391: typo "Jiangfengia"

Resolved

Line 445: typo "(C)" should be (D)

Resolved

Reply to Reviewer #3

A pancrustacean brain and its visual systems define a Cambrian “great appendage” arthropod. by Nicholas J. Strausfeld ^{2*}, Xianguang Hou ¹, and Frank Hirth ^{3*}

This MS presents amazingly detailed comparisons between the neural structures of a Cambrian megacheiran arthropod and those of modern crustaceans. The biological data are all of high quality.

We appreciate that the reviewer approves the quality of the presented biological data. This approval completely contradicts the ensuing criticisms which revolve around the misconception that we claim that traits defining the jianfengioid brain correspond to those of extant Eucrustacea. This was not the case in the submitted ms and is further emphasized in the revised manuscript.

However, most interpretations of the fossil species are based on questionable often very weak fossil evidence (see examples below). The extensive use of image processing (e.g. Photoshop) is sometimes misleading. There is a huge difference between the idealized processed images and reconstructions, and what we actually see in the fossil specimen.

We expect that a reviewer would consult extensive supplementary material for an unbiased assessment of the reported findings. There seems to be a misunderstanding how fossil neural traces are imaged and reconstructed. The revised work now has an expanded analysis of the microscopic neural traces, using four specimens rather than just two. We have employed two strategies for reconstructions: one using human tracing “by eye”; the other using Gaussian amplification of traces comprising dark deposits. It is gratifying that the two strategies converge. These two sets of results are now shown in Figure 3, their neuroanatomical interpretation in Figure 4 and their interpretation in a phylogenetic context in Figure 5. Our detailed description of how Gaussian blur can be utilized in revealing discrete structures is described in the methods section and further detailed in the legend to Fig. 3. This was explained in the original submission, and has been understood by the three other reviewers, according to their commentaries.

Jianfengia is interpreted as a lower stem pancrustacean, based on the assumed structure of its nervous system. However, its appendage structure (head appendages) is completely different from that of pancrustacean. The authors do not take into account these key anatomical aspects.

Neither in the original submission nor in this revision do we claim *Jianfengia* as a lower stem pancrustacean. Our now extended analyses showing correspondences with extant taxa identifies two neuropil arrangements, one malacostracan-like, the other branchiopod-like which may suggest an intermediate evolutionary stage of the *Jianfengioid cerebrum*. This interpretation is further supported by newly added neurocladistics which focus on neural traits and cephalic appendages that resolve *Jianfengia* as sister to total Mandibulata. These correspondences are

now provided in an amended Figure 4. The newly added phylogenetic inferences are presented in new Figure 5 and supplementary Figure S7.

Line 45- What kind of digital adjustments ? This is a key point. Please give details.

The methods applied were described in great detail in the original submission and again here in the results section “Reconstructing the brain of *Jianfengia*” and in the methods section “Isolating neural traces” and in figures 3 and 4, as well as in the supplementary figures.

Line 48- The plural of retina is retinae (Latin)

The conditions for using the word retina does not draw on the latinized plural in the context of studies of visual systems. There the technical plural is ‘retinas’.

Line 61- I agree that the frontal rounded feature is the anterior sclerite (ASC). It appears to be a sclerotized feature comparable with other hard parts of the exoskeleton (see figure B below). However, I see no convincing evidence of an underlying “system of lenses”. Clearly this brownish feature is over-interpreted. I am saying that soft tissues (including sensory organs) do not occur underneath the sclerite but they are not visible in this fossil specimen.

To support the allegation of over-interpretation the reviewer appends blurred out-of-focus copies of small images from our submission . The large, detailed figures available in the supplementary material demonstrates that we interpret, not overinterpret. Hence the reviewer’s comments are at odds with the collective opinions of the three other reviewers. Also, we do not refer to an “underlying system” of lenses: lenses comprise a dioptric system overlying photoreceptors. These are clearly indicated in the enlargement of what is now Figure 2.

Line 62 + “*Their detailed organization suggests structural homology to crustacean nauplius eyes (8, 9) and insect ocelli (10, 11) (Fig. 1C, D; SI Appendix Fig. S1) comprising three lenses overlying a palisade of photoreceptors*” This statement is speculative considering that no detailed organization can be seen in the fossil specimen in question.

Details were provided in the original submission’s supplementary information, which is now Fig. 2. We are confident in our identification of ocelli and their cognate central projections, as are the other reviewers of this work. Nor do we use the word “speculate” but use the word “suggest” indicating reasoned interpretation.

Line 65- Please explain what you mean by superimposition

The Oxford Dictionary’s definition of the word is: “the action of placing or laying one thing over another, typically so that both are still evident.” We note that three other reviewers all agree on our demonstration of ocelli. And following the recommendation of one reviewer who suggested that this is such significant evidence that it ought to be placed in the main body of the paper, we have accordingly done so (now Figure 2).

Line 68- This is perfectly true but what the readers would like to see here (in the “Results” paragraph) are more convincing fossil evidence.

Direct demonstration of gene expression cannot be shown for a fossil. As explained in the original submission and again here, domains of the forebrain and midbrain are defined by the unique combinatorial activity of gene homologs as Character Identity Networks (CHINs). The ChIN underlying ocelli formation and their nerve projections in extant pancrustaceans define the prosocerebrum. Thus, evidence that fossil ocelli send axon projections into the most anterior part of the *Jianfengia* brain suggests the same organization as in extant pancrustacea. It allows the assumption that the prosocerebral genetic program was already in action in the Cambrian (if not much earlier). This also applies to identifying the fossil protocerebrum: we ascribe terminals from the compound eyes as defining its protocerebrum because it receives inputs that underlie the protocerebrally ChIN. This is (and was) explained in the narrative of our paper.

Line 76-77. The lateral eyes are well-preserved. However, this type of eyes occurs in numerous Cambrian arthropods that clearly have no affinities with eumalacostracans (e.g. Artiopoda such as *Acanthomeridion*; also Isoxys, a basal megacheiran or even Tuzoia, that have no close affinities with pancrustaceans)

This comment is factually incorrect. Compound multi-ommatidial eyes are not demonstrated in taxa the reviewer cites. The eLife 2024 preprint of *Acanthomeridion* (doi.org/10.7554/eLife.93113) shows paired eyes, one enlargement of which appears to have lost its entire outer surface: no evidence or mention of facets provided. Searching for demonstrable faceted eyes claimed by the reviewer failed us: *Tuzoia* as well as *Odaraia* have pedunculate eyes; but facets are not demonstrated. Perfectly round profiles are crowded the perimeter of Isoxys eyes; no crustacean-like hexagonal arrangement is demonstrable; there is no evidence of ommatidia. Further, regarding affinities, there’s not yet a phylogenetic analysis of Tuzoia, Isoxys or Acanthomeridion based on neural/sensory characters that would resolve the systematic status of those taxa.

Line 85. The light spots seen in Fig. E are interpreted as possible ommatidia However, similar spots seem to occur elsewhere in the specimen. Unfortunately, I have no access to the high-resolution version of the text-figures. It is important that the authors provide comparative high-resolution images of the lateral eye and other areas of the fossil (see below). I would suggest to provide back-scattered SEM images of the eye. It has been done with the eyes of *Waptia* from the Burgess Shale and clearly revealed the outline of ommatidia that do not appear in light photographs (see Vannier et al. 2018).

Enlarged images of the compound eye were included in the suppl. Data. High-res figures were available with the initial submission and this one, as is the policy of the journal.

Backscatter images in the cited Vannier work show no evidence that they belong to an eye. The images provide dark patches claimed as hexagonally arranged. They do not appear to be which would disqualify them as true compound eye. Nor could they be when none of the published *Waptia* specimens (doi.org/10.1098/rsos.172206) is preserved with a flattened ocular surface.

Backscatter SEM can be quite destructive but is relatively risk free on hard matrices typifying BST preservation. But BSEM is inappropriate for the mudstone Chengjiang matrix. BSEM was originally developed for analyzing crystalline/chemical organization in geological samples. Using it on fossils risks misidentification crystalline deposits as organic traces. The reviewer is thus disingenuous in claiming that ommatidia in *Waptia* do not appear in light (sic) photographs but only in SEM images.

Line 87- I don't think many readers will be convinced by images processed by Photoshop. Indeed, you cannot “*determine if these are taphonomic artifacts or that they represent fossilized cellular elements of the ommatidia*” by using image processing.

Please use other methods to reveal possible underlying retinal structures such as SEM. You may also use a Keyence digital microscope. Although the preservation mode of your specimen is different, SEM studies clearly revealed the inner structure of the eye of Jurassic thylacocephalans (see Vannier et al. in Nat. Comm)

The color of specimens of *Jianfengia* are typical of Chengjiang fossils. Digital color cameras record colors. The “Keyence” microscope cited by the reviewer is a proprietary name for one of several mid-high resolution digital microscopes available at the Yunnan Key Laboratory for Paleobiology (YKLP).

Bar one (Fig. 3f) images of fossilized neuropils in the present work were taken at the YKLP between 2012 and 2018 using instrumentation there that provided color figures shown in eight papers listed below published by NJS and coauthors. In challenging the veracity of images in the present account, the reviewer challenges the veracity of images published in: Nature 2012, 2013, 2014; Phil. Trans R. Soc, 2016; Current Biology 2015, 2016; and Science 2022, 2023.

The reviewer appears to assume that all fossil material is alike. It is not. Those working on Chengjiang biota know that secondary weathering gives rise to coloration and that coloration is often fortuitously constrained to specific organ systems or exoskeletal attributes.

We would not partially destroy fossil material just for using SEM on a substrate that is wholly unsuitable for that technology. Chengjiang mudstone is not equivalent to BST preservation. Nor would we want to split open a jianfengioid eye to emulate Vannier's highly fortuitously fractured *Dolocaris* specimen (Fig. 7: <https://doi.org/10.1038/ncomms10320>).

Line 90 + Fig. 2C is supposed to show the inner structure of assumed ommatidia. Unfortunately these processed images fail to provide evidence that a consistent 4-fold structure occurs in these assumed ommatidia. Honestly, I am not convinced. The outlines of these “light spots” are irregular and blurred. Again, I suggest the authors to use different methods (see above).

Again, one would not try and split open the retina of what is the best fossil of a species for which there are less than a dozen specimens.

Figs D-F are also problematic. The risk with image processing is to make features appear as you want them to appear.

We detail the methods applied in the results section “Reconstructing the brain of *Jianfengia*”; in the methods section “Isolating neural traces”; and in figures 3 and 4, as well as supplementary figures S1 and S4. Suggesting that the authors “to make features appear as you want them to appear” is a comment unworthy a scholarly review.

Line 95 Some spots seem to be more pentaradial than tettraradial (see “3” below)

Out of four visible groups of putative cone cells, one may be pentaradial, another may be septaradial and two tettraradial. Applying Occam’s razor, our simplest interpretation is that what we see are likely tetrads, accounting for two outliers not being coaxial with that of the microscope objective. However, we agree that there may be ambiguities that may be broadly informative, promoting the question whether *Jianfengia* is ancestral to Myriapoda and Pancrustacea. Thus, at this stage of jiangfengioid evolution there was no fixed number of Semper cell prolongations. Toward resolving this (and other aspects of the phylogenetic position of *Jianfengia*) we modified the summary schematics and carried out phylogenetic analyses using primarily neural traits (but not cone cells), following early accounts demonstrating their use for reconstructing brain evolution (doi:10.1098/rspb.2006.3536; also, doi.org/10.1016/j.asd.2011.02.002). This present revision (for data see Supplementary Information) has elucidated the phyletic position of *Jianfengia* as sister to total Mandibulata.

The reconstructions are unfortunately very speculative and based on no solid fossil evidence. What is said about the eyes of modern crustaceans is perfectly correct. The main concern with this MS is the lack of strong fossil evidence.

Our study is based on empirical data provided by extremely small fossils. That is quite an accomplishment in itself (see scale bar in Fig. 1a) and we use all the empirical information that the specimens supply. To claim that the reconstructions are “very speculative” is without merit as is demonstrated in Fig. 3: this is not speculation but methodical analysis.

I really have a problem with these two images. The upper one shows 5 “light areas” but only 4 appear in the lower image. It looks very odd to me.

It is essential to understand the requirement of selecting a narrow band of the available color range that would best transfer to intensity measurement across the image.

Line 107 + I agree that there are ganglia (e.g. represented by dark area in B and F). However, the assumed paired labral ganglia are not visible. What we see in this area is a vague cluster of dark spots that could represent any kind of soft tissues (see Aria et al. in Bioessays). Again, the weakness of this MS is the lack of fossil evidence and the overinterpretation of most features.

This paper does not discuss ganglia. There is evidence of paired labral neuropil: not a pair of ganglia.

Areas and extensions typified by dense contrasting granularities are not mere “vague” clusters. Condensed black deposits in the fossil cephalon comprise material evidence. This is what we must work with and interpret in the light of what is known about the euarthropod cerebrum, its labral neuropils and their evolution in euarthropods. Other studies demonstrate the ground pattern location of the labral neuropils, explaining this feature and its occurrence (see Ortega-Hernández and Wolfe: <https://doi.org/10.1016/j.cub.2021.08.065>).

Please provide details concerning the method used to obtain this image (below). Apparently it was obtained via the juxtaposition of one side of the head with its mirror image. This leads the reader to believe that the structures are indeed perfectly symmetrical and paired. Again, there is a huge difference between this idealized image and what we actually see in the fossil specimen. This way of showing results is unfortunately somewhat (not intentionally) misleading.

The image shown by the reviewer is of the specimen YKLP 11367. The image is asymmetric due to its slight a-p rotation thereby revealing two different levels of traces. It is not “idealized.” Euarthropods are members of Bilateria so how could this image be a mirror image juxtaposition?

Is there any trace of nerves that would belong to the great appendage?

Yes, there are, and these are indicated in Fig. 4a. This question is somewhat surprising considering that the reviewer earlier questioned why great appendages weren’t considered by us in this paper. They appeared as supplementary figure in the original submission (they are shown again here as suppl. Fig. S1) and the possession of great appendages by *Jianfengia* occupied a considerable portion of the discussion as it does again in this revision.

Line 163- The authors go too far in their interpretations (see discussion part) that are actually based on very weak evidence (see my remarks above). I don’t think *Jianfengia* is a lower stem pancrustacean. Please look at its appendages (see other papers; e.g. number and shape of head appendages). They have nothing to do with the appendages of a stem-crustacean (no mandible, etc..)

As stated earlier, both in the original submission and in this revision, we nowhere claim(ed) *Jianfengia* as a lower stem pancrustacean but as a possible evolutionary intermediate with malacostracan-like and other branchiopod-like neuropil arrangements (see revised Figure 4).

We discuss the lack of mandibles and the absence of trunk tagmatization, in terms of the likely underlying genetics and in the context of arthropod evolution. Both are consistent with our interpretation of the reconstructed brain of *Jianfengia* and its evolutionary relationship within Euarthropoda as revealed by a newly added neurocladistics analysis.

FIGURE 5 would really be an excellent one if the authors had sufficient fossil evidence. The comparative drawings of extant species are perfectly correct and very clear.

The reviewer claims that the fossils are incorrect whereas the two extant species are perfectly correct. It is converse: the fossils are correctly depicted but there were two errors regarding gnathal segmentation of the two extant species. These have now been corrected.

Reply to Reviewer #4

Reviewer #4 (Remarks to the Author):

This study analyses four specimens of the species *Jianfengia multisegmentalis* and provides new anatomical information about them thanks to the use of UV fluorescence and enhanced contrast methodology. This has led the authors to describe three main new characteristics of *Jianfengia* neuroanatomy, which establish a link to Pancrustacea, thereby improving our knowledge of pancrustacean neuroanatomy and the inner relationships of megacheirans. The three major findings of the paper are:

We appreciate reviews such as this one that are open, thus obviating anonymity. We also appreciate that this reviewer endorses our use of UV fluorescence and contrast methodology. In reaction to one anonymous reviewer who considered enhanced contrast reconstructions as “idealized overinterpretations” we have included in this revision a reconstruction using the conventional strategy of “tracing” to underline the difference of the conventional method – which is unable to exclude “expectation bias” – with the enhanced method endorsed here, which is virtually “hands-off.”

1. The structural homology of *Jianfengia* anterior sclerite with crustacean nauplius eyes.
2. The presence of Semper cells in the lateral stalked eyes.
3. The nauplius-like eyes of *Jianfengia* and its nervous system are connected only in the forebrain region, in a manner similar to pancrustaceans.

The methodology, though I do not consider myself an expert on UV fluorescence, the results, and their interpretation and discussion are solid. UV optical techniques have been frequently used on similar materials to describe the nervous system and, once again, prove to be effective.

Again, we appreciate the endorsement of the methods used by us.

Out of the three main results described, I have struggled only with identifying the triplet of lenses in the anterior sclerite (see below). However, due to the more easily identifiable radiated structure of the photoreceptors, which I found clear, I do not question the interpretation of the anterior sclerite as homologous to the nauplius eyes. Therefore, I concur with the three major findings of this work.

We are very glad to have your agreement with the main findings. We have now superimposed the outline of three lenses, commenting that they would be the most plausible arrangement with reference to what we know from extant examples of ocellar/nauplius dioptrics.

I also believe that the contribution of this study to our knowledge of early euarthropod

neuroanatomy, and its role in phylogenetic assessment, has the potential to significantly aid in resolving the relationships of major clades and in clarifying megacheiran classification. This contribution is more than significant and is likely to be useful for further studies. As such, I recommend its publication in Nature Communications.

The major revision of this work has been in identifying empirical evidence of neural and cephalic traits that can be used for neurocladistics to resolve evolutionary relationships among stem and crown Euarthropoda. As shown in Figure 5 of this revision, this has revealed the position of two major subgroups of paraphyletic Megacheira: Jianfengiidae as sister to total Mandibulata; and Leanchoiliidae sister to total Chelicerata. This finding raises the interesting possibility that "Megacheira" should no longer be considered a clade but instead the two types of 'great appendages' as convergent character traits of deutocerebral appendages. We address this possibility in the revised discussion of the manuscript in the context of euarthropod evolution.

Nevertheless, I suggest a minor revision to improve the exposition of the results and to enhance the discussion in some areas.

-Lines 33-35: Add citations for Myriapoda, the oldest evidence of neuroanatomy.

The revision puts these elsewhere, but the citations are now present.

-Lines 51-52: I would smooth this phrase and say something like, "Together, these findings suggest Jianfengia as a stem euarthropod closer to pancrustaceans than to both Fuxianhuia and Alalcomenaeus." Again, it is not that I am not convinced of the results, but rather that there is conflicting evidence in other characters.

We wholly agree. And our implementation of neurocladistics (see Fig. 5) together with an amended discussion has, we hope, clarified this .

-Line 57: "barely1mm" should be "barely 1 mm."

OK

-Lines 99-102: Rephrase and possibly split the sentence for better readability.

OK

-Line 391: Missing caption for le.

Corrected

-Figure 5: It would benefit from the addition of a schematic of Fuxianhuia. Also, consider using a hand-drawn cladogram based on your findings and the literature to illustrate the possible evolution of the anatomies depicted.

There are still some open questions regarding whether Fuxianhuiids possessed an intercalary (tritocerebral) ganglion, or whether their caliper-like paired postoral appendages

relate to the mandibular segment T2. Caution suggests that including *Fuxianhuia* in Figure 6 might be premature. However, the outcome of our neurocladistic analysis nevertheless resolves the phylogenetic position of *Fuxianhuia* as sister to Myriapoda within Mandibulata.

-Figure S1: Please provide better contrast for the triplet of lenses, at least in the drawing. The structure is difficult to discern here and in Figure 1. Maybe add schematics to help the reader visualize what you see.

Figure S1 has now been shifted to Fig. 2 with a superimposed outline of what is the most plausible arrangement of the lenses, namely three arranged over a palisade of photoreceptors. The original figure 2 has now been placed in the supplementary data.

-I strongly encourage the author to add a paragraph in the discussion about the possible implications of ontogeny for the anatomy of the nervous system and eyes in *Jianfengia*. Considering the possible life stage of the studied specimens, as well as of the other key taxa, could provide a clearer understanding of the evolutionary forces at play. Since many characters change during ontogeny, we do not always know whether the specimens represent fully developed adults, and character development may occur with shifts in relative timing among different groups. Addressing this aspect could help reconcile some of the conflicting signals observed in different traits (e.g., antennae/great appendage expression) and offer a clearer perspective on the mosaic evolution seen among *Fuxianhuia*, *Alalcomenaeus*, and *Jianfengia* itself.

We agree and have included the question whether the two neuropil arrangements, one malacostracan-like, the other branchiopod-like might suggest an intermediate developmental stage of *Jianfengia*. The (here cited) 2014 paper by Liu et al. on development stages of *Leanchoilia* suggests that at least short-bodied megacheirans likely underwent direct development. We now note that the much rarer *Jianfengia* specimens all suggest identical segmentation and identical dimensions of the carapace/appendages: these imply that the few specimens that exist likely represent at least the same stage of development even if not fully mature. But there is no evidence of "Orsten-like" intermediaries suggesting an anamorphic development nor have – thus far – small versions of *Jianfengia* been described.

I do not wish to see the manuscript again, thus I will be happy to do so if the editors will consider it relevant.

Lorenzo Lustri

Responses to final requests/remarks from reviewer 1,2:

Reviewer #1 (Remarks to the Author):

This has become an excellent paper now. With great respect, we note that following our suggestions, this paper has been rewritten and has been changed in crucial points, as assigning Jianfengia basal, but not within the Mandibulata, because the cone-cell number had not yet evolved within Jianggegiidae. Now Jianfengia's position within Euarthropoda is established through a solid, diligent, and careful cladistic analysis, and in particular neuroanatomical and cephalic traits. The results are absolutely convincing and well exposed. Perhaps one could have more emphasized that Jianfengia obviously has a transitional state between stem and crown-group arthropod, because it bears features of both (number of cone cells vs, brain system). Great work – my congratulation.

We further thank the reviewer for her scholastic assessment and creative suggestions

I found some small points:

Line 32 end: a “t” too much

corrected

Line 92, 93: This organization corresponds to eyestalk morphology typifying eumalacostracan crustaceans and ...

We will find out later (l. 152 ff) why this is mentioned here, but at this point it may seem a bit suggestive. I would suggest to mention that these stalked eyes, as typical of other Cambrian arthropods, also, correspond to those of modern eumalacostracan crustaceans ...

We have modified this to add clarity that may have escaped reviewer. Thus:

“This organization corresponds to eyestalk morphology typifying eumalacostracan crustaceans¹⁵...”

Line 153: Both -> both

The reviewer seems to have mistaken the section heading as the introductory sentence of the section, which indeed starts as Both...

Reviewer #2 (Remarks to the Author):

The authors addressed successfully all questions and concerns of the four reviewers. I already liked the previous version, but after applying the suggestions by different experts I consider this manuscript as more robust, and I would like to see it published in its current form.

Response to reviewer

We thank this reviewer for their strengthened endorsement.

Review

PD Dr. Brigitte Schoenemann

University of Cologne

Dept. of Zoology (Neurobiology/Animal Physiology)

Biocentre, Zùlpicherstrasse 47b

50674 Cologne/Germany

B.Schoenemann@uni-koeln.de

A pancrustacean brain and its visual systems define a Cambrian “great appendage” arthropod.

Nicholas J. Strausfeld, Xianguang Hou, and Frank Hirth

As its predecessors, this manuscript succeeds excellently in revealing fossilized nervous systems, sometimes, as here, in very small organisms and more than half a billion years old. The authors aim to show that coevally lived stem and crown arthropods during the Early Cambrian in the Chengjiang Biota by the example of *Jianfengia multisegmentalis* (Hou, 1987), a not (completely?) tagmatized arthropod. Two more characteristics should be fulfilled to assign *J. multisegmentalis* to pancrustaceans– a tetraconate crystalline cone and a pancrustacean brain. We agree with the nervous system's interpretation but consider the discussion of the eye systems as problematic. Finally, we come to a conclusion, different from the authors, as stated below:

This is an ambitious manuscript that aims to show that Early Cambrian fossils from the Chengjiang Biota reveal recognizable crown groups, namely representatives of tertaconates/panarthropods, already existing coevally with early stem arthropods (e.g. *Fuxianhuia protensa* Hou, 1987 [stem mandibulate], *Alalcomenauis* [stem ch elicerate]) more than half a billion years ago.

The authors describe 3 specimens of the genus *Jianfengia multisegmentalis* (Hou, 1987). To investigate neural traces indicating shapes of brains, ganglia, and neural pathways within a carapace barely 1mm wide may be considered quite a challenge. Despite the absence of trunk tagmatization, the latter would be typical for pancrustacea/tetraconates; in addition, the authors aim to demonstrate typical traits of pancrustaceans in *J. multisegmentalis* to provide a 'stemward' identity towards Pancrustacea/Tetraconata as there are:

- a) Structured eye stalks (-> crown Malacostraca)
- b) Compound eyes with ommatidia, revealing tetradic structures (-> cone-building Semper cells)

- c) nauplius-like eyes and associated nerves supplying a discrete forebrain region -> pancrustaceans
- d) neuropils and tracts in the eye stalks -> nested optic neuropils as occur in eucarustaceans
- e) organization of the deuto- and tritocerebrum -> differentiation from stem Chelicerata and extinct Myriapoda

If successful, the authors would show that neuroanatomical traits could provide a powerful tool for distinguishing characteristics to establish euarthropod relationships, even if external traits that differentiate identities are absent.

This is a difficult review because, of course, the manuscript presents its own interpretation of the anatomical findings. Still, another view in the literature sheds a slightly different light on some aspects. In good academic tradition, I will not attack this manuscript but briefly explain this different perspective. Based on this, I will discuss the results presented in the manuscript, and ultimately, the reader may take his or her position

So, what has been achieved?

a) Structured eye stalks (-> crown Malacostraca)

The joints of the articulated eye stalks can be seen very clearly. The authors indicate one, and there is another more distally (Fig. 1F). There are mentioned 2 (line 78), and an arrow should also indicate the second one.

It is commonly accepted that Malacostraca possess stalked eyes, and at least among the extinct and fossilized I know, the stalks are indeed always articulated. The other way around, I would expect that among the early arthropods, the artiopods, for example, we may find articulated eye stalks. So, I would be slightly more cautious with the conclusion drawn here [Meanwhile, these articulated stalked eyes have been found and described: Schmidt, Schoenemann, et al. 2025, *Biology Communications*, in press]. So probably articulated eye-stalks are typical for crustaceans (not hexapods, belonging to Pancrustacea also!), and consequently, not all arthropods with articulated eye stalks are pancrustaceans. Even the eye stalks of *Leanchoilia* and *Alalcomenaeus* are articulated, see Fig 1 C,D.

[FIGURE REDACTED]

Figure 1 Articulated eye stalks of the arthropod *Pygmaclypeatus daziensis* Zhang, Han & Shu 2000, of *Leancoilia*, and *Alalcomenaeus*,

b) Compound eyes with ommatidia, revealing tetradic structures (-> cone-building Semper cells)

The term semper cells defines cone cells of crustaceans and insects. The manuscript develops the idea of including *J. multisegmentalis* within the stem Pancrustacea. The manuscript should take into account the initial and provisional nature of the use of this term.

The question of the existence of tetradically arranged cone-cells in *J. multisegmentalis* is an important, critical point. Indeed, *J. multisegmentalis* possesses compound eyes, and the loss of cuticular lenses in specimen YKLP 11117 offers insights into the arrangement of internal structures of the ommatidia. Photoprocessing methods were applied to clarify the individual elements. However, if I were to take the position of devil's advocate,

I could say that the threshold can be adjusted until 'it fits' (I am sure the authors haven't done this.)

It is phenomenal that the authors succeed here in showing the cone cells. The problem for me here is that none of the examples clearly shows 4 cone cells; one may find even more at different sizes and numbers (Fig. 2). That the application of photo-processing brings out the tetraconate pattern for me suggests, in conclusion, that there are more than 4 (~6-8) cone-cells, but some of them, four in number, are more pronounced in their development than the others. This may indicate that we here are on a transitional way from a multiconate eye (-> Limulus-like (Xiphosura, Chelicerata), ~ 100 cone cells, Fahrenbach 1968, Harzsch & Hafner 2006) to a tetraconate system. But still, it is not achieved yet.

In the manuscript, the system here is distinguished from the scutigrid myriapods by mentioning one single extension of the cone cell contributing to the secretion of the crystalline cone in tetraconates, while the myriapod possesses two. I am pretty sure one may not find these delicate structures in a fossil. Here, it may help with the argument that the scutigrid ommatidium is developed secondarily from the myriapod ocelli, which in turn originated from original ommatidia (e.g. Paulus 1979, 2000).

Müller et al. (2003) find for the myriapod *Scutigera coleoptrata* (Linnaeus, 1758), 8 cone segments, produced by four crystalline cone cells; tripartite cones (Ascothoracida and Cirripedia), and bipartite ones they suggested to have derived from the tetrapartite system. Thus, four cone cells alone may not be sufficient to postulate an assignment to Tetraconata, even if it is the standard definition. At least this should be discussed more differentiated. It may well be that not all cone cells/segments can be found here in the specimens of *J. multisegmentalis*; in one of the examples presented in the manuscript (Fig. 2,4), there may be evident even 8 elements (D4, another shows up 7, D3), suggesting a tetrapartite origin as in *Scutigera*.

[FIGURE REDACTED]

Figure 2. The cone cells of genus *Jianfengia multisegmentalis* (Hou 1987) (from figure 2 of the manuscript)

c) nauplius-like eyes and associated nerves supplying a discrete forebrain region -> pancrustaceans

It would be great to have them, and I am willing to find them in the figures of the main text; I have tried hard ... I see the filamentous structures, and they may be relicts of a palisade of receptors, and I agree with the pigment. Still, please make the 3 nauplius-like eye elements more evident in your main text figures. I find them in the UV (S1 C). A probable explanation indeed would be that this was one organ consisting of three ocelli

covered by a membrane, which would explain the homogenous appearance of the whole system under white light. I agree with you that the organ may have been movable. The movable character, I think, results from the not-fixed 'stalk,' which clearly protrudes from the carapace. See the dark line (the margin of the carapace, indicated by the arrows in our Fig 2c).

The small element in the reconstruction drawing (main text in Figure 4D) shows the same interpretation as I gave it above and in Fig. C3, which is great, of course, but this finding should be established more clearly in the main text.

If we accept that the filaments are relicts of receptors, please note that they do not show any grouping. Typically, a Nauplius eye is a single median eye and consists of three- or four-pigment cup ocelli. This nicely can be detected in your Figure S1C.

I am not sure about the arthroal membrane here, but I share the idea that this whole structure could have been movable (see above).

I think these results (3 ocelli, movable system) should be more emphasized and shifted from the Supplement to the main text.

So, we agree with a tripartite, nauplius-like eye, probably built by ocelli.

Figure 3. The Nauplius-organ

Neural organization

d) neuropils and tracts in the eye stalks -> nested optic neuropils as occur in eucarustaceans

e) organization of the deuto- and tritocerebrum -> differentiation from stem Chelicerata and extinct Myriapoda

Tracing up the neural structures in this tiny organism, neuropils as pathways is fascinating, very skillful, and convincing. One should never forget that all these investigations start in a quarry, and we deal with 'rocks', fossils. The optic neuropils 1-3 can be discerned very clearly, the reconstruction of the entire brain for example by overlaying the results of different findings and the assumption of a bilateral organism, by that, the methods of reconstruction for completing, for example, by flipping elements of the one side to the other, are justified. The tables (S1, S2) are interesting, and the drawn reconstructions (Fig 5) are excellent and consequent. Unfortunately, the inscriptions in Figure 5 (as perhaps the whole figure) are too small and hardly readable/discernible.

I am not an expert in arthropod brains and the genetic background of the genetic organization of an arthropod segmentation, especially the nervous system, but one can follow the given reasoning, and a possible interpretation seems to have emerged.

In Figure 3 it is not clear what is from YKLP 11367 and YKLP 17299

Discussion

The discussion of the manuscript makes the point that the *J. multisegmentalis* among the stem-arthropods of the Chengjiang biota represents a true pancrustacean, because, although a trunk tagmatization still is missing, the eyes have a tetraconate ommatidium, *J. multisegmentalis* possesses as typical for pancrustaceans a nauplius-like eye consisting of three elements, and a brain that is organized as in pancrustaceans.

We may agree with the latter, but we see a problem with the interpretation of the eye systems.

Note that median eyes (-> Nauplius-like eyes, Nauplius eyes = Median eyes of crustaceans) also exist in the megacheiran *Leanchoilia!* (Garcia-Bellido & Collins 2007, Schoenemann & Clarkson 2012, 2023), and indeed in *Alalcomenaeus* (see below).

[FIGURE REDACTED]

Figure 4 Median eyes of *Alalcomanaeus cambricus* Simonetta, 1970 and *Leanchoilia superlata* Walcott 1912

This has implications and should be considered and discussed.

Even being aware of the work of Tanaka *et al.* 2013, one may be a bit careful with the interpretation of the eyes of *Alalcomanaeus* and *Leanchoilia*. The specimens of *Alalcomanaeus* illustrated by Briggs & Collins (Fig 4.1, 4.4) or of Whittington (1981, Fig. 126) clearly show that *Alalcomanaeus* has club-shaped, pedunculate compound eyes (here Fig. 1D), surmounting articulated stalks; the same is true for *Leanchoilia* (here Figure 1 B, C). Here, the eyes and their stalks are much finer.

Note that there are median eyes in the megacheiran *Leanchoilia* (e.g. Garcia- Bellido & Collins, 2007; Schoenemann Clarkson 2012) and in *Alalcomenaeus* also (e.g. Briggs & Collins, 2007, Fig. 4.3), as can be seen clearly here in Figure 4. This has implications and should be considered and discussed - for example, consequently, *Jianfengia* is NOT the only Early Cambrian genus identified with median eyes (and crustacean-like eyestalks). Perhaps *J. multisegmentalis* is among the first with 3 median eyes. Still, there are crustaceans with 4 ocelli in their median eye complex, e.g. among the branchiopoda (e.g., Reimann & Richter 2007) – so the number alone may is not game-changing here.

We agree, however, completely with the idea that genetically based the morphological variance of homolog deutocerebral appendages is wide enough to comprise great appendages, antennules formed as 'pinchers' or pier-like antennules, or ...

In general, we also agree with the interpretations of the neural systems and the homologies explained here. I am not an expert in arthropod nervous systems, but I think the deductions are reasonable.

Finally, one of this manuscript's great conclusions is that not the Hox-gene-associated divergences of the segmented trunk define ancestral and crown arthropods, but do the segmental cerebral domains, the unique lineage-specific neuropils, and associated traits. This is an important point, but our conclusion differs slightly from what the authors indicate here.

Even if one of the characteristics (like those of the nervous system as here) has passed the 'finish-line' to reach a crown group state, while others have not (tagmatization of the body), one has to be aware that as here, obviously boundaries of the stem-/crown groups cannot always be rigidly defined, and that there are transitional situations (as represented in the eye-status of *Jianfengia multisegmentalis* (Hou 1987)). Here, one part (brain system) has reached that stage, and the other (tagmatization of the body) still has not; the eye system of *Jianfengia multisegmentalis* (Hou 1987) is on its way. This may have important implications for describing evolutionary processes in general.

The results here show that the formation of different crown-group characteristics (tagmatization of the trunk – crown-group-nervous-system – eye system) may evolve with different velocities. An important point because one now will have to weigh when the complete, or a 'sufficient' crown-group-state has been arrived. Probably it is not adequate if only one of the characteristics has passed the 'finish line'. Even if the focal point in the considerations of this manuscript has been laid on the eyes and brain/nervous structures, it may be an essential fact that the tagmatization of the body is not quite there yet.

Nevertheless, the authors are right to state that neuroanatomical traits could provide a powerful tool for distinguishing characteristics to establish euarthropod relationships, even if external traits that differentiate identities are absent.

References

Butterfield, N. J. (2002). *Leancoilia* guts and the interpretation of three-dimensional structures in Burgess Shale-type fossils. *Paleobiology*, 28, 155-171.

Briggs, D.E.G. & Collins, D. 1999: The arthropod *Alalcomanaeus cambricus* Simonetta, from the middle Cambrian Burgess Shale of British Columbia. *Palaeontology* 42, 953–957.

Fahrenbach, W. H. (1968). The morphology of the eyes of *Limulus*: II. Ommatidia of the compound eye. *Zeitschrift für Zellforschung und Mikroskopische Anatomie* 93, 451-483.

García-Bellido, D. C., & Collins, D. (2007). Reassessment of the genus *Leancoilia* (Arthropoda, Arachnomorpha) from the Middle Cambrian Burgess Shale, British Columbia, Canada. *Palaeontology*, 50, 693-709.

Harzsch, S., & Hafner, G. (2006). Evolution of eye development in arthropods: phylogenetic aspects. *Arthropod Structure & Development*, 35, 319-340.

Hou, X., & Bergström, J. (1997). Arthropods of the Lower Cambrian Chengjiang fauna, southwest China. *Fossils and Strata* 45, 1-177.

Müller, C. H., Rosenberg, J., Richter, S., & Meyer-Rochow, V. B. (2003). The compound eye of *Scutigera coleoptrata* (Linnaeus, 1758)(Chilopoda: Notostigmophora): an ultrastructural reinvestigation that adds support to the Mandibulata concept. *Zoomorphology*, 122, 191-209.

Paulus, H. F. (1979) Eye structure and the monophyly of the arthropod eye. In *Arthropod Phylogeny* (ed. Grypta, A. P.) 299–383 (Nordstrand, 1979).

Paulus, H. F. (2000). Phylogeny of the Myriapoda–Crustacea–Insecta: a new attempt using photoreceptor structure. *Journal of Zoological Systematics and Evolutionary Research*, 38, 189-208.

Reimann, A., & Richter, S. (2007). The nauplius eye complex in 'conchostracans' (Crustacea, Branchiopoda: Laevicaudata, Spinicaudata, Cyclestherida) and its phylogenetic implications. *Arthropod Structure & Development*, 36, 408-419.

Richter, S., Loesel, R., Purschke, G., Schmidt-Rhaesa, A., Scholtz, G., Stach, T., ... & Harzsch, S. (2010). Invertebrate neurophylogeny: suggested terms and definitions for a neuroanatomical glossary. *Frontiers in Zoology*, 7, 1-49.

Schmidt, M*, Schoenemann, B.*, Hou X.-g., Melzer & Liu, Y. (2025) A unique 1 Lower Cambrian arthropod with two different compound eye systems. *Communications Biology* (2025), in press * shared first authorship

Schoenemann, B., & Clarkson, E. N. (2012). The eyes of *Leancoilia*. *Lethaia*, 45, 524-531.

Schoenemann, B., & Clarkson, E. N. K. (2023). The median eyes of trilobites. *Scientific Reports* 13, 3917.

Tanaka, G., Hou, X., Ma, X., Edgecombe, G. D., & Strausfeld, N. J. (2013). Chelicerate neural ground pattern in a Cambrian great appendage arthropod. *Nature*, 502(7471), 364-367.

Whittington H. B. (1981). Rare arthropods from the Burgess Shale, Middle Cambrian, British Columbia. *Philosophical Transactions of the Royal Society, Series B*, 292, 329–357.

A pancrustacean brain and its visual systems define a Cambrian “great appendage” arthropod.

by Nicholas J. Strausfeld ^{2*}, Xianguang Hou ¹, and Frank Hirth ^{3*}

This MS presents amazingly detailed comparisons between the neural structures of a Cambrian megacheiran arthropod and those of modern crustaceans. The biological data are all of high quality. However, most interpretations of the fossil species are based on questionable often very weak fossil evidence (see examples below). The extensive use of image processing (e.g. Photoshop) is sometimes misleading. There is a huge difference between the idealized processed images and reconstructions, and what we actually see in the fossil specimen.

Jianfengia is interpreted as a lower stem pancrustacean, based on the assumed structure of its nervous system. However, its appendage structure (head appendages) is completely different from that of pancrustacean. The authors do not take into account these key anatomical aspects.

Line 45- What kind of digital adjustments ? This is a key point. Please give details.

Line 48- The plural of retina is retinae (Latin)

Line 61- I agree that the frontal rounded feature is the anterior sclerite (ASC). It appears to be a sclerotized feature comparable with other hard parts of the exoskeleton (see figure B below). However, I see no convincing evidence of an underlying “system of lenses”. Clearly this brownish feature is over-interpreted. I am saying that soft tissues (including sensory organs) do not occur underneath the sclerite but they are not visible in this fossil specimen.

Line 62 + “Their detailed organization suggests structural homology to crustacean nauplius eyes (8, 9) and insect ocelli (10, 11) (Fig. 1C, D; SI Appendix Fig. S1) comprising three lenses overlying a palisade of photoreceptors”

This statement is speculative considering that no detailed organization can be seen in the fossil specimen in question.

Line 65- Please explain what you mean by superimposition

Line 68- This is perfectly true but what the readers would like to see here (in the “Results” paragraph) are more convincing fossil evidence.

Line 76-77. The lateral eyes are well-preserved. However, this type of eyes occurs in numerous Cambrian arthropods that clearly have no affinities with eumalacostracans (e.g. Artiopoda such as *Acanthomeridion*; also *Isoxys*, a basal megacheiran or even *Tuzoia*, that have no close affinities with pancrustaceans)

Line 85. The light spots seen in Fig. E are interpreted as possible ommatidia

However, similar spots seem to occur elsewhere in the specimen. Unfortunately, I have no access to the high-resolution version of the text-figures. It is important that the authors provide comparative high-resolution images of the lateral eye and other areas of the fossil (see below). I would suggest to provide back-scattered SEM images of the eye. It has been done with the eyes of *Waptia* from the Burgess Shale and clearly revealed the outline of ommatidia that do not appear in light photographs (see Vannier et al. 2018)

Line 87- I don't think many readers will be convinced by images processed by Photoshop. Indeed, you cannot "*determine if these are taphonomic artifacts or that they represent fossilized cellular elements of the ommatidia*" by using image processing. Please use other methods to reveal possible underlying retinal structures such as SEM. You may also use a Keyence digital microscope. Although the preservation mode of your specimen is different, SEM studies clearly revealed the inner structure of the eye of Jurassic thylacocephalans (see Vannier et al. in Nat. Comm)

Line 90 +

Fig. 2C is supposed to show the inner structure of assumed ommatidia. Unfortunately, these processed images fail to provide evidence that a consistent 4-fold structure occurs in these assumed ommatidia. Honestly, I am not convinced. The outlines of these "light spots" are irregular and blurred. Again, I suggest the authors to use different methods (see above).

Figs D-F are also problematic. The risk with image processing is to make features appear as you want them to appear.

Line 95

Some spots seem to be more pentaradial than tetraradial (see "3" below)

The reconstructions are unfortunately very speculative and based on no solid fossil evidence. What is said about the eyes of modern crustaceans is perfectly correct. The main concern with this MS is the lack of strong fossil evidence.

I really have a problem with these two images. The upper one shows 5 “light areas” but only 4 appear in the lower image. It looks very odd to me.

Line 107 + I agree that there are ganglia (e.g. represented by dark area in B and F). However, the assumed paired labral ganglia are not visible. What we see in this area is a vague cluster of dark spots that could represent any kind of soft tissues (see Aria et al. in Bioessays). Again, the weakness of this MS is the lack of fossil evidence and the overinterpretation of most features.

Please provide details concerning the method used to obtain this image (below). Apparently it was obtained via the juxtaposition of one side of the head with its mirror image. This leads the reader to believe that the structures are indeed perfectly symmetrical and paired. Again, there is a huge difference between this idealized image and what we actually see in the fossil specimen. This way of showing results is unfortunately somewhat (not intentionally) misleading.

Is there any trace of nerves that would belong to the great appendage ?

Line 163- The authors go too far in their interpretations (see discussion part) that are actually based on very weak evidence (see my remarks above).

I don't think Jianfengia is a lower stem pancrustacean. Please look at its appendages (see other papers; e.g. number and shape of head appendages). They have nothing to do with the appendages of a stem-crustacean (no mandible, etc..)

FIGURE 5 would really be an excellent one if the authors had sufficient fossil evidence. The comparative drawings of extant species are perfectly correct and very clear.